# L-lysine protects C2C12 myotubes and 3T3-L1 adipocytes against high glucose damages and stresses

**S. Mehdi Ebrahimi[1], S. Zahra Bathaie⊙[1]\*, Nassim Faridi[1], Mohammad Taghikhani[1†], Manouchehr Nakhjavani[2], Soghrat Faghihzadeh[3]**

**1** Department of Clinical Biochemistry, Faculty of Medical Sciences, Tarbiat Modares University, Tehran, Iran, **2** Endocrinology and Metabolism Research Center (EMRC), Vali-Asr Hospital, School of Medicine, Tehran University of Medical Sciences, Tehran, Iran, **3** Department of Statistics, Zanjan University of Medical Sciences, Zanjan, Iran

† Deceased.
\* bathai_z@modares.ac.ir, zbatha2000@yahoo.com

**Data Availability Statement:** All relevant data are within the manuscript and Supporting Information files.

**Funding:** The Research Council of Tarbiat Modares University paid for preparation of materials in this

## Abstract

Hyperglycemia is a hallmark of diabetes, which is associated with protein glycation and misfolding, impaired cell metabolism and altered signaling pathways result in endoplasmic reticulum stress (ERS). We previously showed that L-lysine (Lys) inhibits the nonenzymatic glycation of proteins, and protects diabetic rats and type 2 diabetic patients against diabetic complications. Here, we studied some molecular aspects of the Lys protective role in high glucose (HG)-induced toxicity in C2C12 myotubes and 3T3-L1 adipocytes. C2C12 and 3T3-L1 cell lines were differentiated into myotubes and adipocytes, respectively. Then, they were incubated with normal or high glucose (HG) concentrations in the absence/presence of Lys (1 mM). To investigate the role of HG and/or Lys on cell apoptosis, oxidative status, unfolded protein response (UPR) and autophagy, we used the MTT assay and flow cytometry, spectrophotometry and fluorometry, RT-PCR and Western blotting, respectively. In both cell lines, HG significantly reduced cell viability and induced apoptosis, accompanying with the significant increase in reactive oxygen species (ROS) and nitric oxide (NO). Furthermore, the spliced form of X-box binding protein 1 (XBP1), at both mRNA and protein levels, the phosphorylated eukaryotic translation initiation factor 2α ($p$-eIf2α), and the Light chain 3 (LC3)II/LC3I ratio was also significantly increased. Lys alone had no significant effects on most of these parameters; but, treatment with HG plus Lys returned them all to, or close to, the normal values. The results indicated the protective role of Lys against glucotoxicity induced by HG in C2C12 myotubes and 3T3-L1 adipocytes.

## Introduction

Hyperglycemia is a major hallmark of diabetes mellitus and exerts deleterious or toxic effects *in vivo* and on different cell types. High glucose (HG) concentration results in the glucotoxicity that characterized by dysfunction of the cells and eventually ends to diabetes complications

project. It had no role in study design, data collection and analysis, decision to publish, or preparation of the manuscript.

**Competing interests:** The authors have declared that no competing interests exist.

[1]. The circulating HG not only alters vascular endothelial smooth muscle cells (VSMCs) [2], but also other cell types, including skeletal muscle cells and adipocytes [3, 4]. In the normal situation, insulin stimulates glucose (Glc) uptake by both muscle and fat cells [5], the process that was distorted in the absence of insulin or in insulin resistance.

HG can trigger a series of cellular responses and induces direct damage to biological macromolecules including lipids, proteins, and DNA [6]. The nonenzymatic glycation of proteins induced by HG alters protein folding and function, and has been known as the main cause of protein misfolding and damage, which is the source of many diabetes complications [7, 8]. On the other hand, the accumulation of the misfolded proteins in the cells induces endoplasmic reticulum stress (ERS) and unfolded protein response (UPR), which are involved in the development of several conditions including diabetes complications [9–11]. HG can also activate autophagy via ERS signaling [12, 13] that may subsequently lead to the autophagic-induced apoptosis.

The eukaryotic initiation factor $2\alpha$ (eIF2$\alpha$) has been known as the important sensor of the ERS. In response to environmental stresses, a family of protein kinases phosphorylate eIF2$\alpha$ to alleviate cellular injury or alternatively induce apoptosis. This process has been introduced as one of the three arms of the UPR [14]. Another arm of the UPR is XBP1 splicing, which is also activated as a cellular response to ERS [15]. Activation of both of these arms mainly led to the cell apoptosis.

Previous studies have confirmed the destructive roles of reactive oxygen and nitrogen species (superoxide anion/ROS and peroxynitrite, respectively) in tissues and cells under HG condition [16]. Overproduction of nitric oxide (NO) in the presence of superoxide, rapidly forms peroxynitrite, which is a very strong oxidant and has been detected in heart, kidney, nerve and retina of diabetic subjects [17].

Chemical chaperones are small molecules that protect proteins against different types of stresses (such as glycation), stabilize and conserve protein structure, and inhibit protein aggregation [18]. Amino acids are one of the main families of chemical chaperones. L-lysine (Lys), arginine, proline and glycine are the major amino acids in the chemical chaperone family. The free amino group(s) in the amino acids acts as a chemical decoy for reducing sugars [19–21]. So that, in the HG conditions, the free amino acids bind to Glc and competitively inhibit the reaction between Glc and the free amino groups in the side chain or at the N-terminal of proteins. In this case, they prevent protein misfolding and/or unfolding. Lys with a long side chain and two free amino groups shows no toxicity in the rat up to 5% w/w in oral intake [22], inhibits the nonenzymatic glycation of proteins, protects the protein structure and conserves the folding of many proteins in both *in vitro* experiments [19, 23–25] and in the rat model of diabetes [19, 26]. Lys treatment has also shown some improvement against complications in type 2 diabetic patients [24, 25]. Continuing with our previous studies regarding the study of the toxic effects of HG in different biological processes and the protective role of chemical chaperones (especially Lys), here we investigated the effect of Lys on HG-induced stresses in C2C12 myotubes and 3T3-L1 adipocytes. Therefore, in addition to HG-induced oxidative stress and cell death, the effect of Lys on some markers of the UPR (phosphorylation of eIF2$\alpha$ and splicing of XBP1) and autophagy (Light chain 3 (LC3) accumulation) in HG condition were investigated.

## Materials and methods

### Materials

The C2C12 and 3T3-L1 cell lines were purchased from the Iranian Biological Resource Center (IBRC, Tehran, Iran). Penicillin-streptomycin (C-A4122) was purchased from Biosera

(Biosera, UK). Bovine serum albumin (BSA) (A2153), insulin (I1882), Lys (L8662), Oil Red O (O0625), dexamethasone, 3-isobutyl-1-methylxanthine (IBMX) (I7018), RIPA buffer (R0278), and protease inhibitor (P8340) were purchased from Sigma Chem. Co., St. Louis, USA. Glucose (Glc) (K1665937) was bought from Merck (Darmstadt, Germany). Fetal bovine serum (FBS) (S0115) was purchased from Biochrom, Berlin. Dulbecco's modified Eagle's medium (DMEM/low (1803391X) and HG (H1674879)) were purchased from Life Technology, (Gibco), CA. Annexin V-FITC-PI apoptosis detection kit (BMS500FI-100) was purchased from eBioscience (San Diego, US). Horse serum was bought from Veterinary Medicine Faculty, University of Tehran, Iran. The biotin-HRP-labeled anti-rabbit IgG as a secondary antibody were purchased from Kermanshah University of Medical Sciences. The antibody against XBP1s (sc-7160) was from Santa Cruz, USA; and the antibodies against p-eIF2α (phospho-Ser51) (ab32157), eIF2α (ab5369), β-actin (ab227387), and LC3 (48394) were bought from Abcam Inc., Cambridge, MA. ECL plus Western blotting kit (RPN2232) and polyvinylidene difluoride membrane (PVDF) (3010040001) were bought from Amersham Biosciences, Stockholm, Sweden. All other materials and reagents were of analytical grade.

## Cell culture

Mouse skeletal myoblast cell line C2C12 was seeded in 60-mm-diameter culture dishes and grown in 5.5 mM Glc as a normal glucose (NG) or 25 mM Glc as a HG medium in DMEM supplemented with 10% FBS, 1% penicillin 50 U /streptomycin 50 μg/ml (pen/strep) and 2 mM L-glutamine in a 5% $CO_2$ humidified incubator at 37˚C ($CO_2$ incubator). The NG and HG conditions for 3T3-L1 cells were 25 and 50 mM Glc, respectively.

## Cell differentiation induction

To induce differentiation of C2C12 from myoblasts to myotubes, when the cells were 60% confluent, the growth medium was switched to differentiation medium, which was containing 2% horse serum supplemented with insulin [27]. After 48 h of differentiation, the medium was changed every day up to 6 days. After that, the creatine kinase (CK) activity was determined in the C2C12 myoblasts to confirm their differentiation into myotubes [28]. After 6 days, cells were washed twice with calcium- and magnesium-free, ice-cold phosphate-buffered saline (PBS), scraped from the plates into 1 ml PBS and centrifuged at 16,000 × g for 10 min at 4˚C and subjected to the enzymatic assay for determination of the CK specific activity.

The CK activity was measured in units using reagents from Biovision (USA Cat: K777). Then, the total protein in each sample was determined by the Bradford method [29], and the CK specific activity was reported as Unit/μg total protein.

To induce the differentiation of preadipocyte cell line, 3T3-L1, to adipocytes, the cells were grown in 6-well plate in DMEM containing the same components as above. After reaching the confluence, they were extensively rinsed and adipocyte cell differentiation was induced by changing the medium to DMEM containing 10% FBS, 1% pen/strep, 0.5 mM methyl isobutyl xanthine (IBMX), 0.25 μM dexamethasone, 100 μM indomethacin, and 1 μg/ml insulin, for 2 to 3 weeks. Intracellular lipid accumulation due to the progression of the cell differentiation was microscopically assessed using Oil Red O staining, by the method was described by Ramírez-Zacarías et al. [30].

## Effect of HG and/or Lys treatment on the cell viability

For investigating the effect of HG/ Lys on the cell viability, the differentiated cells, C2C12 and 3T3-L1 were plated in the 96-well at $2.5 \times 10^4$ cells/well exposed to either NG or HG with

different concentrations of Lys (0.5, 1, 2, 5, and 10 mM), and incubated up to 48 h. Then, the viability of the cells was evaluated by MTT assay [31].

Briefly, after 48 h, the medium was carefully removed; and then, 200 µl of medium containing 20 µl MTT (5 mg/ml) was added to each well. The cells were then incubated for 4 hours at 37˚C. Then, the medium of wells was removed and 100 µl DMSO was added to each well. Finally, the color intensity of formazan solution was monitored by Multi-Mode Microplate Reader (Cytation™3, Biotek, USA) at 570 nm. In this experiment, the control cells were assumed 100% viable and the viability of the treated cells was calculated relative to the control and expressed as the percentages (%) of viable cells.

Then, other sets of experiments were designed to investigate the cytotoxicity of HG, Lys (1 mM) plus HG (HG+Lys) in comparison with the control group in NG at different time intervals of 0, 2, 4 and 6 h of incubation, and various parameters were evaluated in these cells.

## Apoptosis detection by flow cytometry

Annexin V-FITC apoptosis detection kit was used to detect the apoptosis in the cells at different conditions. The treated cells were finally trypsinized and pelleted by centrifugation at room temperature, 1200 rpm for 5 min, and then, they were washed twice with cold PBS. Cell suspensions were incubated with Annexin V- FITC and propidium iodide (PI) for 15 min at room temperature in the dark and analyzed by BD FACSCantoTM II flow cytometer (USA). The obtained raw results were analyzed by *FlowJo* software version 7.6 and expressed as the percentage of cells undergoing apoptosis.

## Measuring the ROS and NO production

Total intracellular ROS was determined by the permeable fluorescent probe, 70-dichlorohydrofluorescein diacetate (CM-H2DCFDA) [32, 33]. The CM-H2DCFDA dye in the presence of ROS converts to fluorescent DCF, which can be considered as a measure of ROS. The excitation and emission wavelengths of DCF were 490 and 520 nm, respectively [32, 33]. The fluorescence intensity (FI) of DCF in the cells was read and expressed as arbitrary unit (AU).

NO production in the mentioned cells was determined by Sigma-Aldrich 23479 Nitrate/ nitrite Assay Kit Colorimetric (St Louis, USA) in both cell lines, in the presence of HG and/ or Lys. Total NO ($NO_2^-$ + $NO_3^-$) was determined using Griess reagent that produces the azo compound with maximum absorption at 570 nm. The standard curves were then plotted using different concentrations of nitrite and nitrate in the similar reagent and at the same wavelength. Then, the concentration of total NO was calculated by interpolating of the sample absorbency using the linear standard curve.

## Total mRNA extraction and RT-PCR

Total RNA extraction was performed from the Lys treated C2C12 myotubes and 3T3-L1 adipocytes using the protocol supplied with the RiboExTM kit (GeneAll®, Korea) according to the manufacturer's instructions. Total RNA (2 µg) was reverse transcribed (RT) with HyperscriptTM RT Mastermix Kit (GeneAll, Korea). The final PCR products were electrophoresed on 1–1.5% agarose gels containing EB along with DNA Ladder (SMOBIO DM2300 ladder). The expression and splicing of *XBP1* mRNA was determined using RT-PCR. The cDNA sample was amplified using Taq DNA Polymerase Master Mix, Red (Amplicon) in the presence of the XBP1 mRNA primer pair (`forward primer: TTACGAGAGAAAACTCATGGCC and reverse primer: GGGTCCAAGTTGTCCAGAATGC`). Briefly, the reaction conditions consisted of 2 µl of cDNA and 0.2 µM primers in a final volume of 20 µl of master mix. Each cycle consisted of denaturation at 95˚C for 15 s, annealing at 60˚C for 5 s and extension at

72˚C for 10 s, respectively. Amplified fragments covering flanking exon fragments consist s*XBP*1 (spliced *XBP*1) and u*XBP*1 (unspliced *XBP*1) were separated on 1–1.5% agarose gels containing EB along with DNA Ladder (SMOBIO DM2300 ladder). The measurements were performed by three independent experiments in triplicate. The intensity of the PCR product bands was assessed in the *Image J* software (NIH approved) and semiquantitative data were obtained.

### Western blot analysis

The expression levels of *p*-eIf2α, XBP1 and LC3 in the mentioned cells at different conditions and times was investigated by Western blotting.

Cells were rinsed with PBS and then solubilized in a lysis buffer containing a complete cocktail of protease inhibitors. In order to remove any insoluble material and clarify cell lysate, the lysate was immediately centrifuged at 13000 ×g for 10 min at 4˚C, the supernatants were collected and stored at -80˚C. The total sample protein content was determined by the Bradford protein assay. Equal amounts of crude protein homogenates (30 μg) from whole-cell extracts were combined with Laemmli sample buffer and fractionated by SDS-PAGE (8–15%). After electrophoresis, the proteins were transferred to a PVDF for Western blot analysis. Membranes were blocked with 5% BSA in Tris-buffered saline containing 0.1% Tween-20 (TBST). After that, the membranes were incubated with the primary antibodies diluted in TBST and 3% bovine serum albumin overnight at 4˚C. Then, the membranes were washed three times in TBST and incubated with the appropriate concentration of secondary antibody coupled with horseradish peroxidase diluted in TBST with a 5% BSA (1:10,000) for 1 h, at room temperature. After an additional three times washing, immune complexes were visualized through autoradiography using ECL. The films were scanned with an image scanner using *LabScan* software and quantified with the *Image J* analysis software.

### Statistical analysis

Data were indicated as mean ± SD of at least three independent repeats. Statistical analysis was carried out using the repeated-measures ANOVA followed by Tukey's Post Hoc analysis between different groups, and at different time intervals. The P-value <0.05 was considered as statistically significant value. The statistical analysis of the data between groups was presented in the results section and showing by stars in the figures. The within group analyses at different time intervals and the p-values, if statistically significant, were also shown in the figures.

## Results

### Differentiated phenotypes of C2C12 myotube and 3T3L1 adipocyte

As shown in Fig 1, the CK specific activity was significantly (*p* = 0.027) increased in C2C12 cells after six-day incubation in the differentiation media. S1A and S1B Fig show the C2C12 cells before and after differentiation, respectively.

Oil Red O staining was used to show the differentiation of the 3T3-L1 cells. Before differentiation there was no or very rare lipid droplet in the cells. Thus, the cells are colorless (figure not shown). However, as S2A and S2B Fig show, after two- to three-week incubation in the induction media and differentiation, a significant increase in the accumulation of lipid droplets was observed in the cells.

### Lys attenuates the HG-induced cell death

Fig 2A shows the results of the MTT assay in both differentiated cells treated with Lys, which indicates no significant toxicity of different Lys concentrations (0.5 to 10 mM) against both

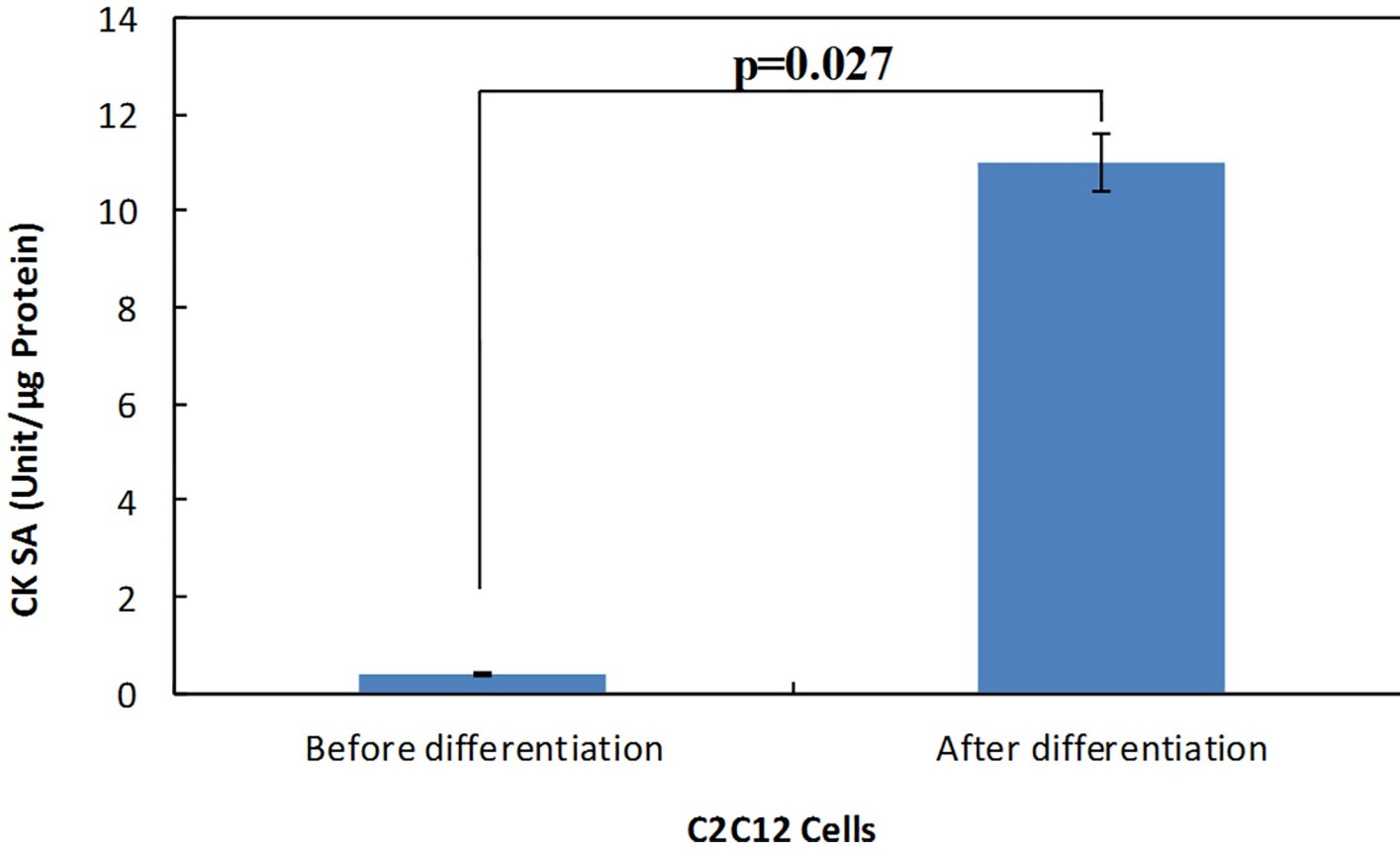

**Fig 1. Biochemical assessment of the C2C12 differentiation.** The creatine kinase specific activity in the cell lysates at first and after 6 days of incubation in the differentiation media. The data was statistically analyzed by t-test and the *P*-value was shown in the figure.

C2C12 myotubes and 3T3-L1 adipocytes. However, avoiding any other toxicity, we used the mild Lys concentration (1 mM) for further experiments, in both cell lines.

Fig 2B and 2C show the effect of HG and HG+Lys, respectively, on viability of both C2C12 myotubes and 3T3-L1 adipocytes at different time courses (2, 4 and 6 h). These figures indicate a significant decrease ($p = 0.000$) in the cell viability due to the HG treatment in comparison with the control cells. Lys treatment significantly overcomes the HG toxicity. So that, the significant differences in the cell viabilities were observed between groups treated with Lys+HG and groups treated with HG in C2C12 myotubes ($p = 0.034$) and in 3T3-L1 adipocytes ($p = 0.038$). Despite the improvement effect of Lys, the viability of the cells treated with Lys +HG was significantly ($p = 0.000$) lower than control cells, in both cell lines. The statistical differences in cell viability within the groups at different time intervals in both cell types are shown in these figures.

## HG induces and Lys inhibits the apoptotic cell death

To demonstrate the apoptosis induction in the cells due to HG and in the presence of both HG +Lys, we applied Annexin V and PI staining using Flow Cytometry. Fig 3A and 3B show the percentages of C2C12 and 3T3-L1 cells, respectively, at each quarter. So that, the population and percentages of alive cells (both Annexin V and PI negative, left bottom), at early apoptosis (positive for Annexin V and negative for PI, left upper), late apoptosis (negative for Annexin V and positive PI, right upper), and necrosis (both Annexin V and /PI positive, right bottom) are

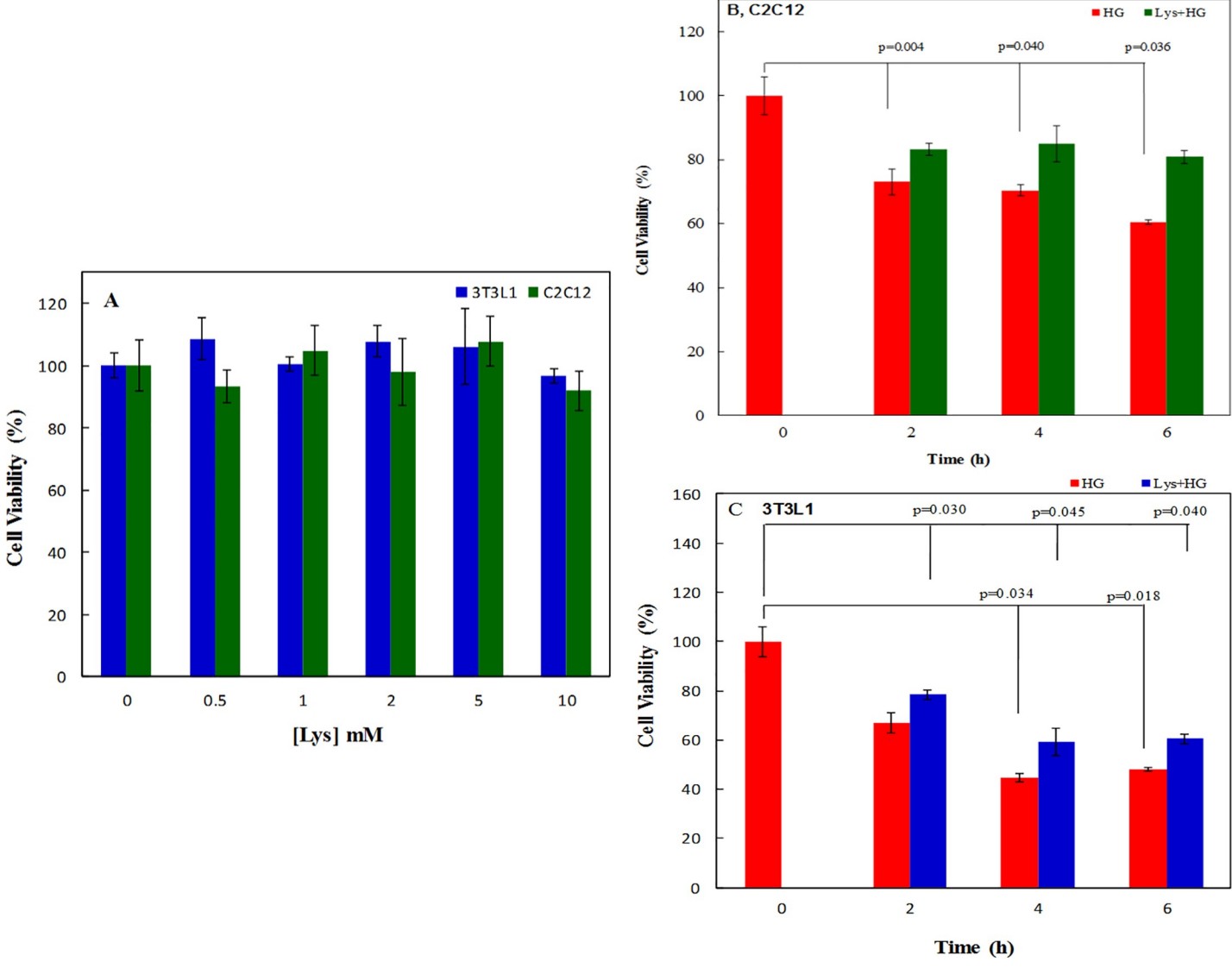

**Fig 2. The *in vitro* viability assay of the cells in different media.** (A) Graphical representation of the MTT assay of C2C12 myoblasts and 3T3L-1 adipocytes after 24 hours incubation with 0.5 to 10 mM Lys. As the figure indicated, there is no toxicity of different concentrations of Lys against these cells. Thus Lys 1 mM was used in all other experiments. (B) and (C) show the effect of HG and HG+Lys on cell viability at different time intervals of 2, 4 and 6 hours, which was determined by MTT assay in C2C12 myotubes and 3T3L-1 adipocytes, respectively. Time zero, 0, was considered as control.

shown at each condition. The numeric estimates of the alive and apoptotic (the sum of both early and late apoptosis) cells after 6 h incubation are tabulated in Table 1. As the data indicate, the cell viability of both differentiated C2C12 and 3T3-L1 cells were significantly ($p = 0.000$) decreased due to the exposure to HG. However, Lys treatment attenuates significantly ($p = 0.000$) both early and late apoptosis in both cell types, after 6 h.

## Inhibitory effect of Lys on ROS and NO production

Fig 4A and 4B show the fluorescence intensity (FI) of DCF as a measure of ROS production in both cell lines in the medium containing HG in the presence or absence of Lys. The statisticl analysis shows significant increase ($p = 0.000$) in the ROS production in HG groups in comparison with the controls of both cell types. Due to Lys treatment, significant decrease ($p =$

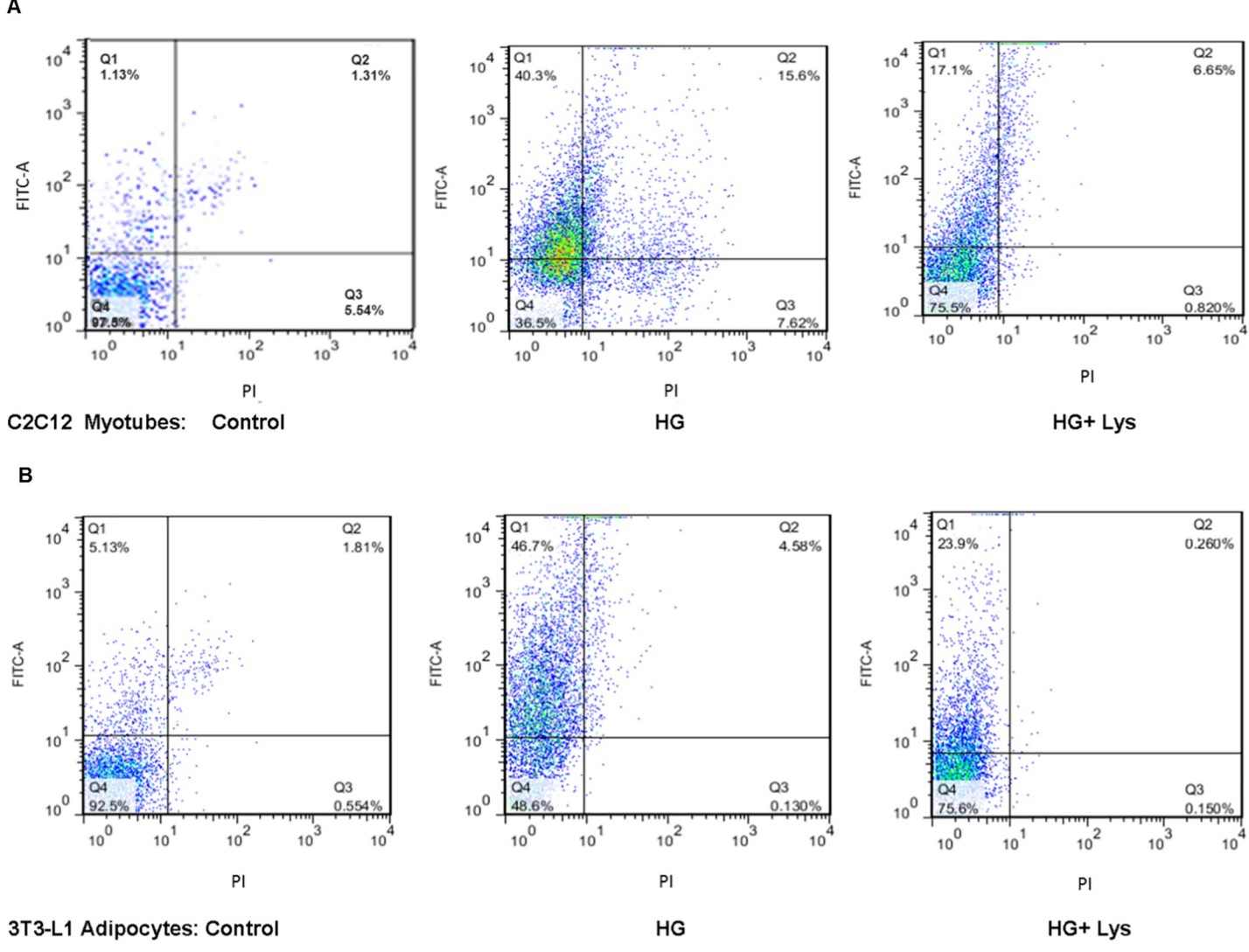

**Fig 3. The panel of the flow cytometry data of the cells in different conditions.** Q1, early apoptotic cells (annexin V$^+$/PI$^-$); Q2, late apoptotic cells (annexin V$^+$/PI$^+$); Q3, necrotic cells (annexin V$^-$/PI$^+$) and Q4, live cells (annexin V$^-$/PI$^-$). (A) The upper figures show the flow cytometry results of C2C12 myotubes and (B) the bottom show the data of 3T3-L1 adipocytes after 6 h incubation in the medium containing normal Glc (Control), high Glc (HG) and both HG and Lys (HG+Lys). The control cells are shown at the left, the cells treated with HG and HG+Lys are shown in the middle and right, respectively. The percentages of the cells in each state are shown in the figures, and the overall data of the alive and apoptotic cells (early+late) are shown in Table 1.

**Table 1. The flow cytometry data of C2C12 myotubes and 3T3-L1 adipocytes after 6 h incubation in different conditions.**

|  |  | Living Cells | (Early + Late) Apoptotic Cells |
|---|---|---|---|
| C2C12 | Control | 92.5 | 6.94 |
|  | HG | 48.6 | 51.28 |
|  | HG + Lys | 75.6 | 24.16 |
| 3T3-L1 | Control | 97.5 | 1.44 |
|  | HG | 36.5 | 55.9 |
|  | HG + Lys | 75.5 | 23.75 |

The statistical differences between all groups were significant, $p = 0.000$.

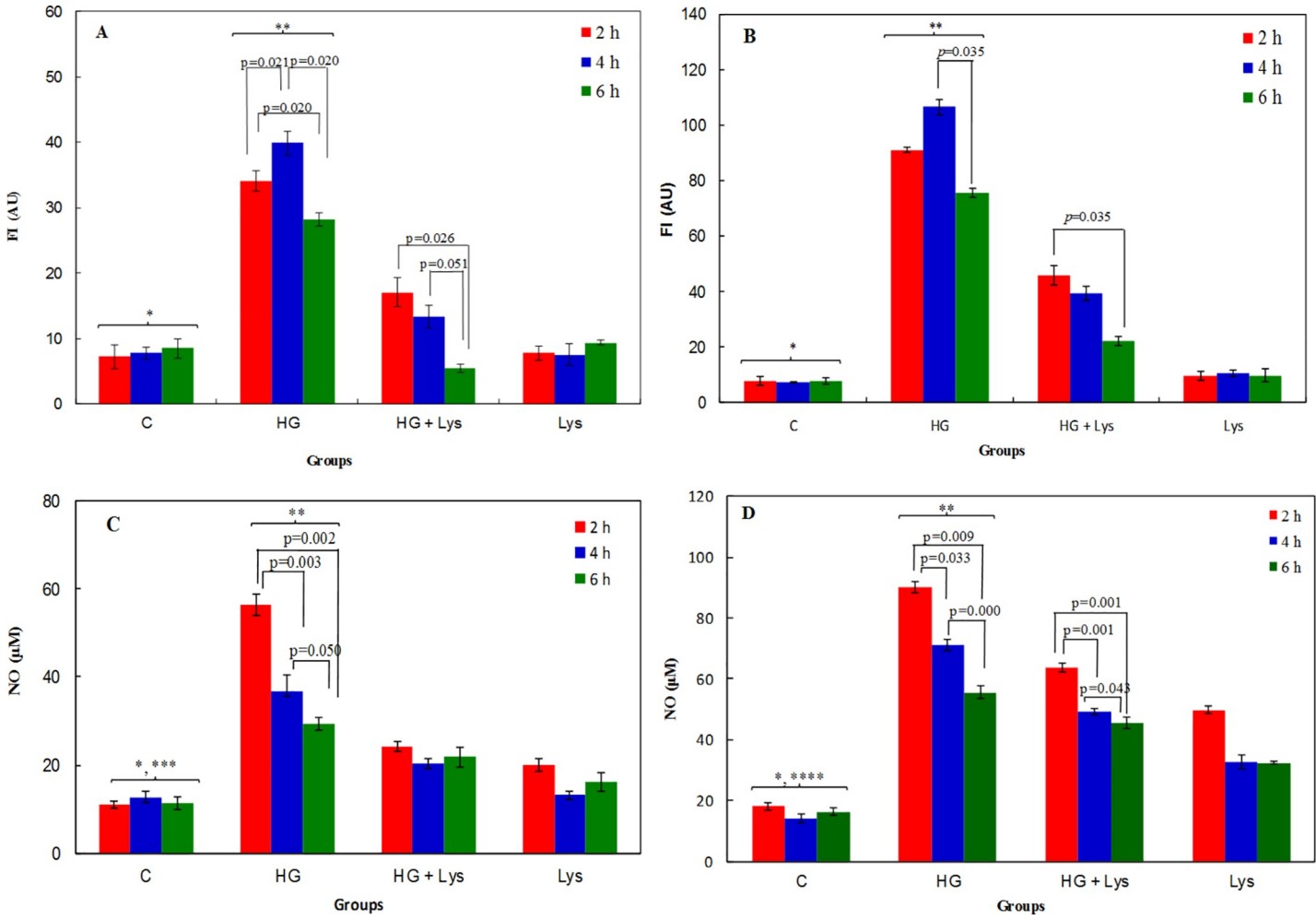

**Fig 4. The oxidative stress markers.** The fluorescence intensity (FI) of DCF as arbitrary units (AU) are shown in (A) C2C12 myotubes and (B) 3T3-L1 adipocytes in the control cells (normal Glc = NG), and in the presence of HG or HG+Lys after 2, 4 and 6 h of incubation. The total NO concentrations are shown in (C) C2C12 myotubes and (D) 3T3-L1 adipocytes, after 2, 4 and 6 h of incubation in different conditions. The maximum NO was produced after 2 h of HG treatment in both cells. The p-values of the differences within groups, at different time intervals, are shown in the figures. The statistical differences between the groups are shown by stars in the figures and defined as follows: * indicates the differences between control and HG groups ($p = 0.000$). ** indicates the differences between HG and HG+ Lys groups ($p = 0.000$). *** indicates the differences between control and Lys groups ($p = 0.009$). **** indicates the differences between control and Lys groups ($p = 0.000$).

0.000) in the ROS levels was observed in HG+Lys groups in comparison with HG groups in both C2C12 myotubes and 3T3-L1 adipocytes.

Fig 4C and 4D show the effect of HG and /or Lys on NO production in C2C12 and 3T3-L1 cells, respectively. In 3T3-L1 adipocytes, a statistical analysis shows the significant differences ($p = 0.000$) between the NO in all four groups (control, HG, HG+Lys and Lys); however, the magnitudes of the NO were higher in the HG group than the others and due to the treatment with Lys (in HG+Lys groups) it was decreased significantly. In C2C12 myotubes the differences in the NO between control, HG and HG+Lys groups were significant ($p = 0.000$). Lys increased significantly ($p = 0.009$) the NO level in C2C12 myotubes, too. These changes were also time dependent and the maximum value in each group was observed after 2 h of treatment. The statistical differences within groups at different time intervals are shown in the figures.

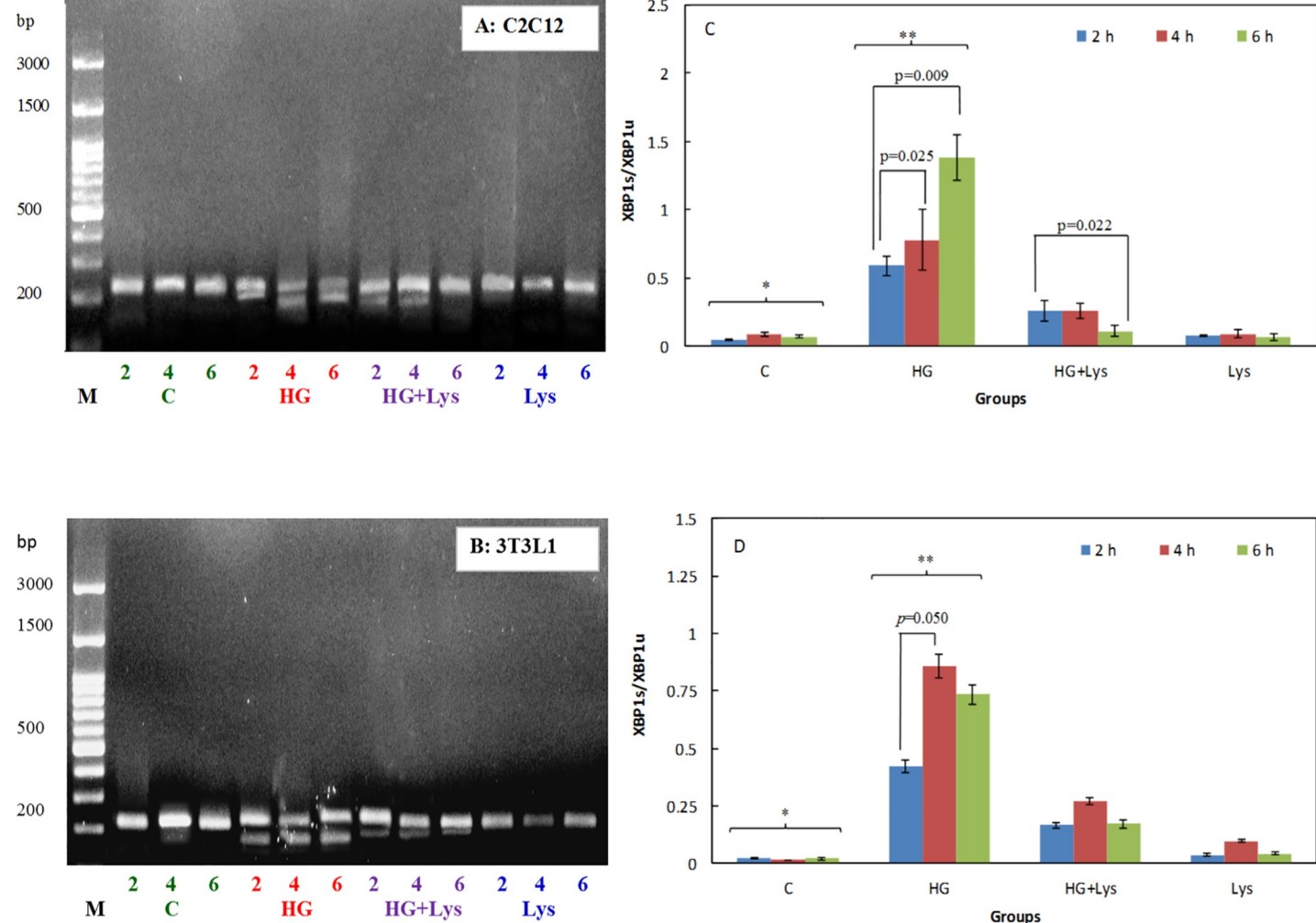

**Fig 5. RT-PCR analysis of XBP1 mRNA splicing in different conditions.** The gel electrophoresis pattern of the spliced and unspliced (XBP-1s and XBP-1u, respectively) amplicons as studied by RT-PCR in (A) C2C12 myoblasts and (B) 3T3-L1 adipocytes with no treatment (control), in the presence of HG or HG+Lys after 2, 4 and 6 h of incubation. The semiquantitative analysis of the XBP1s and XBP1u mRNA values, obtained by ImageJ analysis, and the ratio of XBP1s/XBP1u mRNA are shown in figures (C) C2C12 myoblasts and (D) 3T3L1 adipocytes. The data represent as mean ± SD of three independent experiments. The data of the named four independent groups at different time intervals were analyzed by repeated-measures ANOVA. The significant differences (p-value) in the XBP1 mRNA splicing within the groups at different time intervals are shown in the figures. The statistical differences between the groups are shown by stars in the figures and defined as follows: * indicates the differences between control and HG groups ($p = 0.000$). ** indicates the differences between HG and HG+ Lys groups ($p = 0.000$).

## Lys suppresses ERS response

Some UPR markers were investigated in this study. XBP1 splicing was studied at two levels of mRNA and protein expression. The agarose gel electrophoresis of the final PCR products of each cell line, Fig 5A and 5B, have obviously indicated the increasing amount of the spliced form of XBP1 (XBP1s) mRNA below the unspliced one (XBP1u) by increasing time from 2 to 6 hours of incubation of the cells in HG. Semiquantitative analysis of these bands using *Image J* software and the obtained ratios of the spliced XBP1/ unspliced XBP1 (XBP1s/ XBP1u) mRNA were shown in Fig 5C and 5D for C2C12 myotubes and 3T3-L1 adipocytes, respectively. The analysis of the data in both cell types, using repeated-measures ANOVA, indicated the significant ($p = 0.000$) increase in the XBP1s in HG groups in comparison with the control.

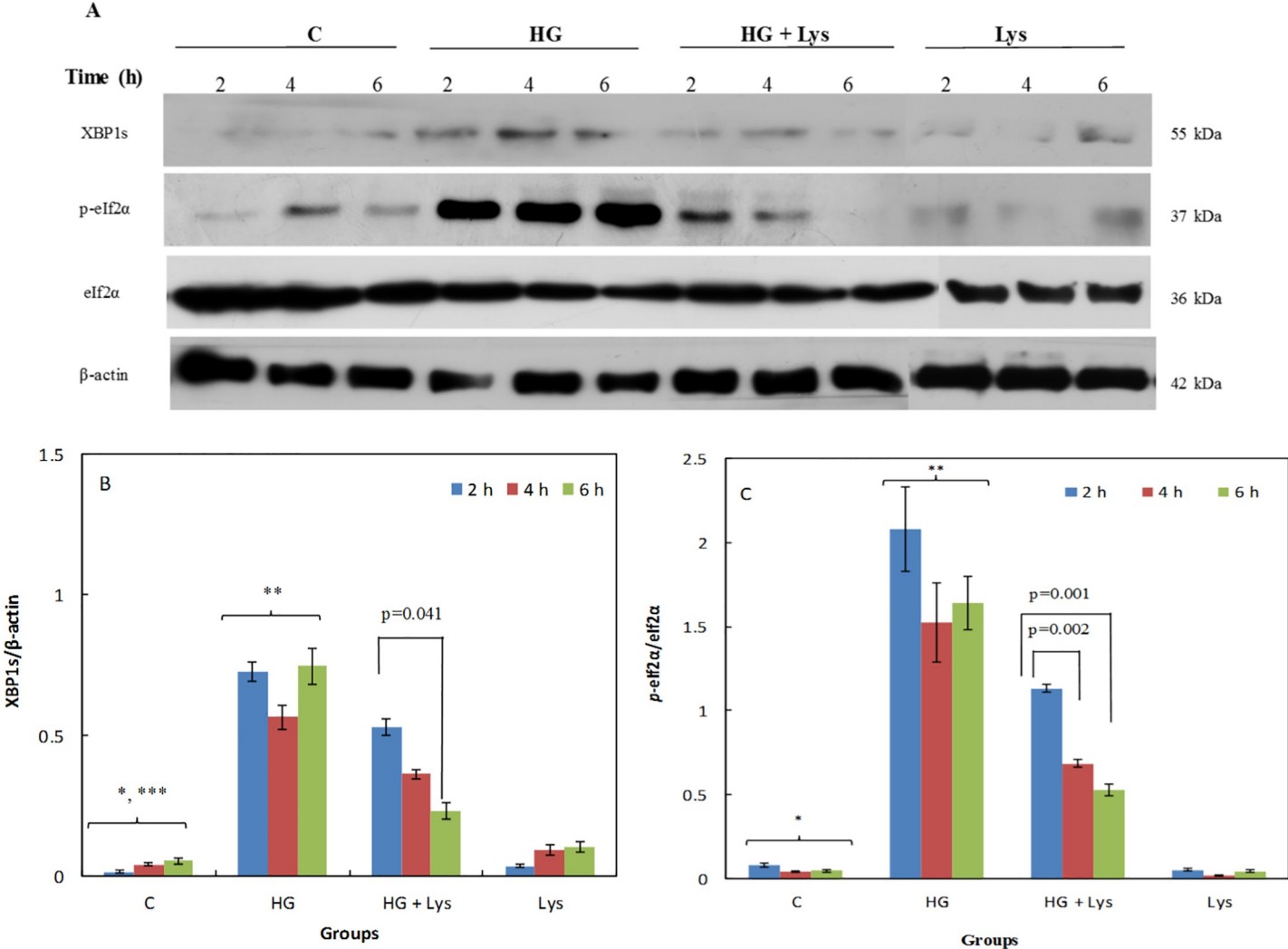

**Fig 6. Western blotting data of the stress responses of C2C12 myoblasts in different conditions.** (A) The Western blot data of whole-cell lysates of C2C12 myoblasts subjected to Lys (1 mM) and/ or HG treatments at different times of incubation that exposed to different antibodies. The β-actin served as a loading control. (B and C) Show the ratio of the spliced XBP1 (XBP1s)/β-actin and $p$-eIF2α/eIF2α, respectively, as determined by semiquantitative analysis of the bands in (A). All data are represented as means ± SD of three independent experiments. The data of the named four independent groups at different time intervals were analyzed by repeated-measures ANOVA. The significant differences (p-value) in the XBP1s/β-Actin and $p$-eIF2α/eIF2α ratios within the groups at different time intervals are shown in the figures. The statistical differences between the groups are shown by stars in the figures and defined as follows: * indicates the differences between control and HG groups ($p = 0.000$). ** indicates the differences between HG and HG+ Lys groups ($p = 0.000$). *** indicates the differences between control and Lys groups ($p = 0.032$).

This value (the spliced form of XBP1 mRNA) was significantly decreased ($p = 0.000$) after Lys treatment in the HG+Lys groups of both cell types.

The Western blot results of two ERS markers using the specific antibodies against the mentioned proteins are shown in Fig 6A (C2C12 myotubes) and 7A (3T3-L1 adipocytes).

Figs 6B and 7B show the XBP1s/β-Actin ratios in the mentioned cells, at different times of incubation. These results also confirm the obtained data of XBP1 mRNA splicing. It means that the ratios of the spliced forms of XBP1 protein to β-actin were significantly increased ($p = 0.000$) after HG treatment in comparison with the control groups in both cells. However, this parameter was significantly ($p = 0.000$) decreased due to the treatment with Lys in the HG +Lys groups in both cells. There was a slight difference between the control and Lys groups in the C2C12 ($p = 0.032$), but this difference was not significant in 3T3-L1.

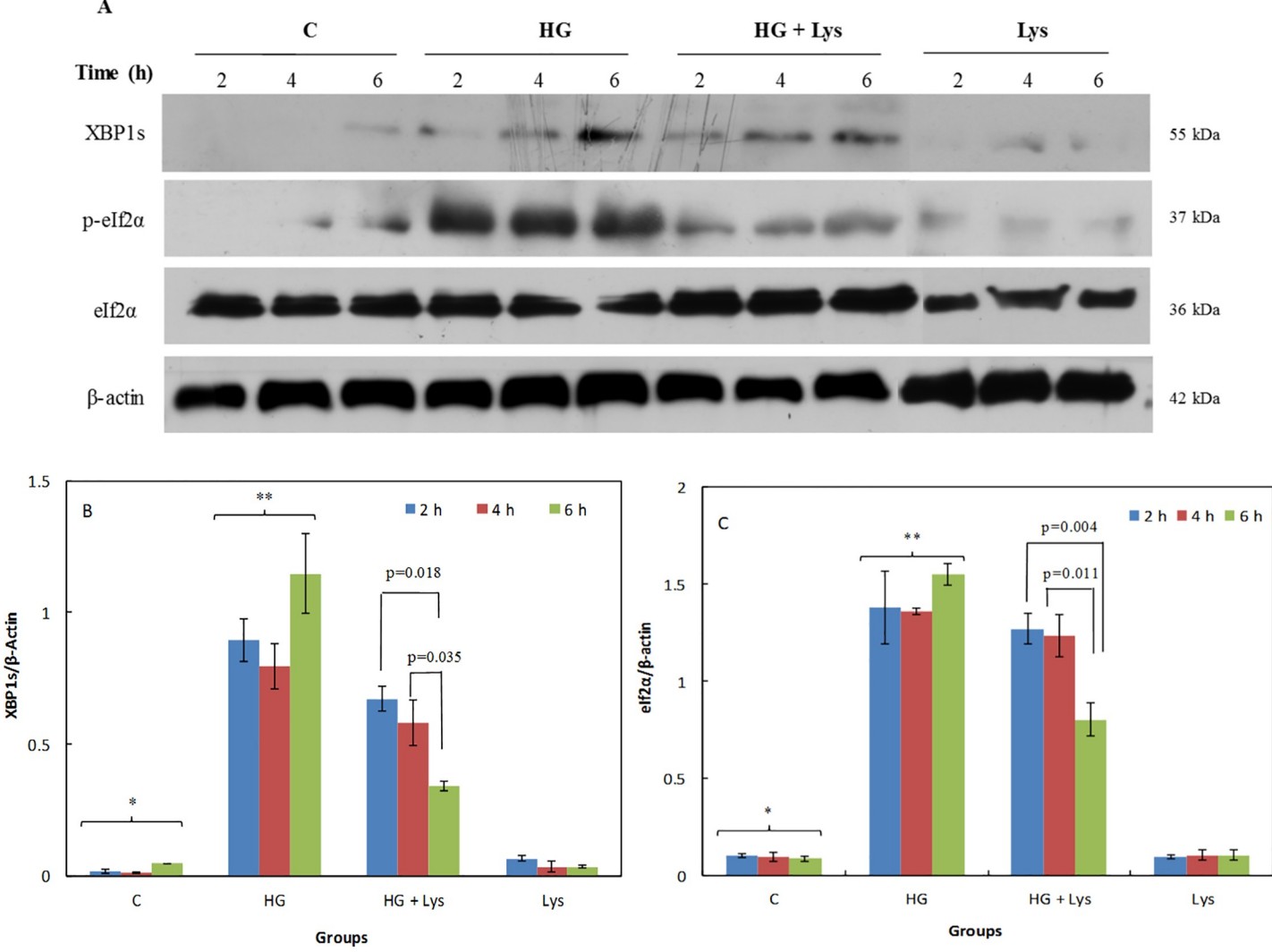

**Fig 7. Western blotting data of the stress responses of 3T3-L1 adipocytes in different conditions.** (A) The Western blot data of whole-cell lysates of 3T3-L1 adipocytes subjected to Lys (1 mM) and/ or HG treatments at different times of incubation that exposed to different antibodies. The β-actin served as a loading control. (B) and (C) Show the ratio of the spliced XBP1 (XBP1s)/ β-actin and $p$-eIF2α/eIF2α, respectively, as determined by semiquantitative analysis of the bands in (A). All data are represented as means ± SD of three independent experiments. The data of the named four independent groups at different time intervals were analyzed by repeated-measures ANOVA. The significant differences (p-value) in the XBP1s/β-Actin and $p$-eIF2α/eIF2α ratios within the groups at different time intervals are shown in the figures. The statistical differences between the groups are shown by stars in the figures and defined as follows: * indicates the differences between control and HG groups ($p = 0.000$). ** indicates the differences between HG and HG+ Lys groups ($p = 0.000$ for XBP1/β-Actin and $p = 0.003$ for p-eIF2α/eIF2α ratios).

Another marker of the UPR is $p$-eIF2α. Fig 6C shows the ratio of $p$-eIF2α/eIF2α in C2C12 myotubes at different conditions and time intervals. Fig 7C shows the results of the same parameters in 3T3-L1 adipocytes. All of these data indicated a significant increase ($p = 0.000$) in the eIF2α phosphorylation at Ser51 in the presence of HG in comparison with the NG in the control groups, in both cell types. Lys treatment significantly ($p = 0.000$ and $p = 0.003$) reversed these changes in the groups treated with HG+Lys in comparison with HG in C2C12 myotubes and 3T3-L1 adipocytes, respectively. There were no significant differences between control and Lys groups of both cell types.

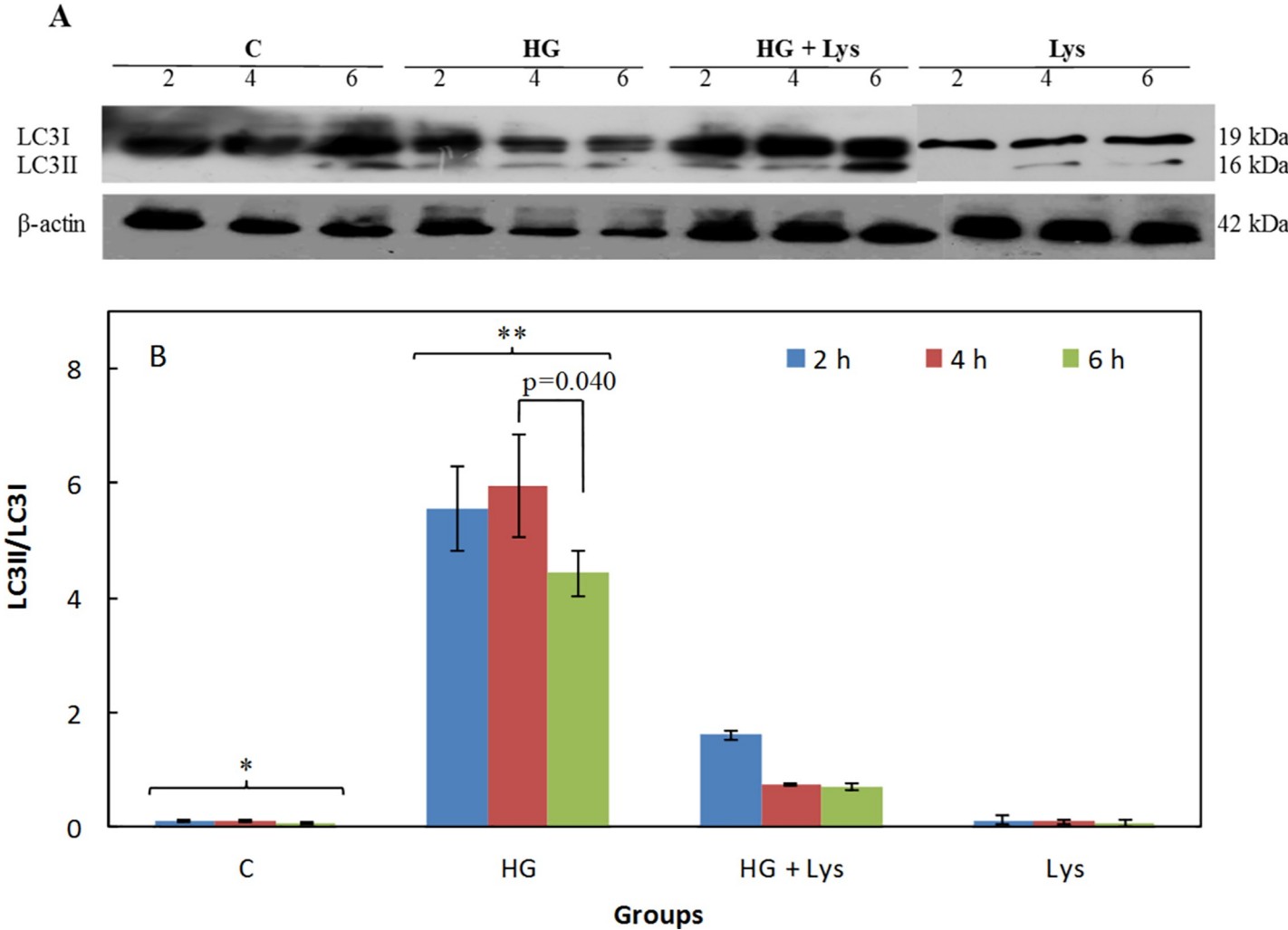

**Fig 8. Western blotting data of the autophagy marker in C2C12 myotubes in different conditions.** (A) The Western blot data of whole-cell lysates of C2C12 myotubes subjected to Lys (1 mM) and/ or HG treatments at different times of incubation that exposed to the LC3 antibody. (B) Shows the ratios of the LC3II/LC3I at different conditions, as determined by semiquantitative analysis of the bands in (A). All data are represented as means ± SD of three independent experiments. The data of the named four independent groups at different time intervals were analyzed by repeated-measures ANOVA. The significant differences in the LC3II/LC3I ratio within groups, at different time intervals are shown in the figure. The statistical differences between the groups are shown by stars in the figures and defined as follows: * indicates the differences between control and HG groups ($p = 0.000$). ** indicates the differences between HG and HG+ Lys groups ($p = 0.000$).

## Lys inhibits HG-induced autophagy

The effect of HG and/or Lys treatment on the expression of LC3I and LC3II is shown in Figs 8A and 9A in the C2C12 myotubes and 3T3-L1 adipocytes, respectively. The LC3-II/LC3I ratios at different conditions are also shown in Figs 8B and 9B, in the mentioned cell types. These data indicate a significant increase ($p = 0.000$) in this ratio due to the HG treatment in comparison with the untreated group, it means the LC3II accumulation. These changes were significantly ($p = 0.000$) reversed by Lys treatment in the HG+Lys groups in both cell types. However, the LC3-II/LC3I ratios were still higher in the HG+Lys treated groups ($p = 0.000$ and $p = 0.006$) than the control groups in C2C12 and 3T3-L1, respectively. Lys alone had no significant effect on this parameter in the mentioned cell types.

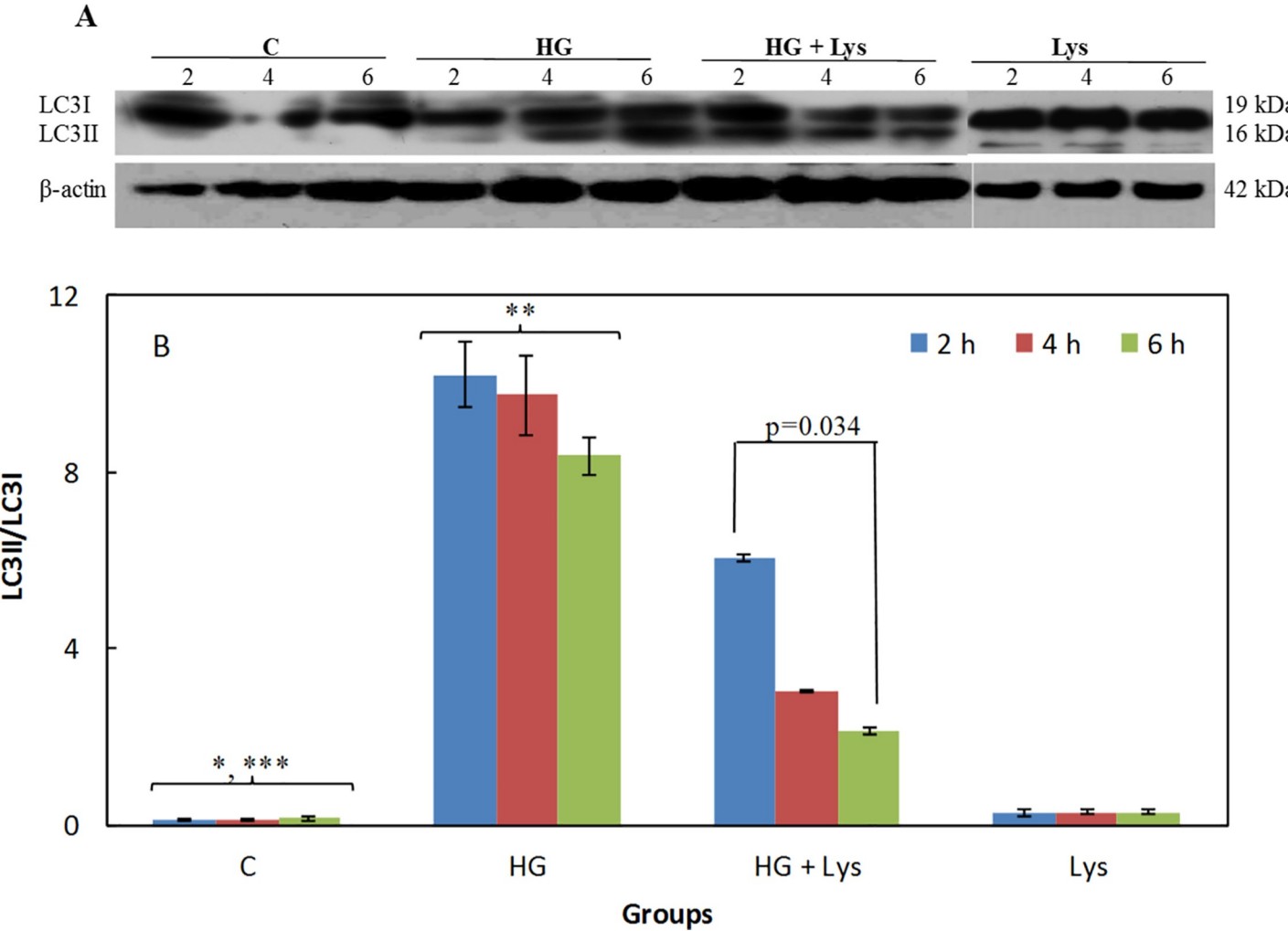

**Fig 9. Western blotting data of the autophagy marker in 3T3-L1 adipocytes in different conditions.** (A) The Western blot data of whole-cell lysates of 3T3-L1 adipocytes subjected to Lys (1 mM) and/ or HG treatments at different times of incubation that exposed to the LC3 antibody. (B) Shows the ratios of the LC3II/LC3I at different conditions, as determined by semiquantitative analysis of the bands in (A). All data are represented as means ± SD of three independent experiments. The data of the named four independent groups at different time intervals were analyzed by repeated-measures ANOVA. The significant differences in the LC3II/LC3I ratio within groups, at different time intervals are shown in the figure. The statistical differences between the groups are shown by stars in the figures and defined as follows: * indicates the differences between control and HG groups ($p = 0.000$). ** indicates the differences between HG and HG+ Lys groups ($p = 0.000$). *** indicates the differences between control and HG+Lys groups ($p = 0.006$).

## Discussion

We found here that Lys prevents the HG-induced cell death in both C2C12 myotubes and 3T3-L1 adipocytes. The beneficial effects of Lys in these cells were through the reduction of oxidative stress and regulation of the UPR and autophagy.

In the present study, we have been applying different concentrations of Lys from 0.5 to 10 mM in both C2C12 myotubes and 3T3-L1 adipocytes. Physiologic concentration of Lys has been reported about 201 to 253 μM according to the type of food intake [34] or 92 ± 6 μM [35]. We used Lys above physiologic concentration and there was no toxicity against the studied cells. However, a previous study indicated the toxicity of high Lys concentration (10 mM) against HK-2 kidney cells [36]. Therefore, we applied the mild Lys concentration (1 mM) in further experiments to avoid any possible toxicity.

The importance of oxidative stress in the pathogenesis of diabetes and diabetic complications have been extensively studied for years and all the data based on animal models and diabetic patients indicated the role of oxidative stress in this disease. Chronic hyperglycemia has been known as a major source of oxidative stress and *de novo* free radical generation, which in order depletes the activity of the antioxidative defense system [37, 38]. Our results show the direct inductive effect of HG on ROS production and the protective role of Lys on this parameter, in both C2C12 myotubes and 3T3-L1 adipocytes.

Another important mechanism causing cell dysfunction in diabetes is related to NO production. It is produced due to the enzymatic reaction catalyzed by three different nitric oxide synthases (NOSs). Among these isozymes, the inducible NOS (iNOS) is known to be involved in production of the most NO and $O_2^-$ in diabetes [39–41]. In addition, the role of liver NO at the early stage of diabetes and its importance in aging has been reported [42]. The presence of high concentration of NO and some ROS products, such as the superoxide radical ($O_2^-$), may lead to the production of another highly reactive oxidant species, peroxynitrite ($ONOO^-$), resulted into more aggressive oxidative and nitrosative stresses [43].

The role of Lys as a selective inhibitor of iNOS has been introduced in the endotoxic shock [44]. We have also shown the anti-diabetic and antioxidant activities of Lys in animal model and human [19, 24–26]. Here, we searched about the effect Lys treatment on the two mentioned oxidative parameters and two important pathways in the *in vitro* models of diabetes.

At first, the role of HG on the cell viability and cell death was investigated. The MTT assay indicated the HG-induced cell death, and the flow cytometry results showed the apoptosis induction in both cell lines. These processes were mediated by a significant increase in the ROS and total NO levels in both C2C12 myotubes and 3T3L adipocytes due to the exposure to HG. The maximum ROS and NO levels were observed at 4 and 2 h of cells exposure, respectively. Although both ROS and NO levels decreased by increasing the incubation time in both differentiated cell lines, they were still significantly higher than the control. Lys alone had no significant effect on ROS production, but significantly increased NO production, especially in 3T3-L1 adipocytes. When Lys co-treated with HG (Lys+ HG), the ROS and NO levels were significantly lower than the HG alone. It means that Lys significantly prevented the ROS and NO production in both cell lines. These results are compatible with our previous data about the antioxidant activity of Lys in diabetic rats [19] and in acute pancreatitis in mice [45]. Furthermore, these results are completely compatible with that reported previously about the role of Lys as an inhibitor of L-arginine uptake and NO synthesis in rat [44].

As mentioned in the introduction, our previous *in vitro* and *in vivo* studies indicated that chronic hyperglycemia affects the structure and function of many proteins, including extracellular proteins (such as fibrinogen [24] and lysozyme [23, 25]), intracellular proteins, including cytosolic proteins (hemoglobin) [19] and even, nuclear proteins (histone H1 [46]), as well as molecular chaperones (α-crystallin) [26]. Other studies indicated that membrane proteins also affected by this harmful condition [47, 48]. The quality control systems in the ER are responsible for the detection of misfolded proteins and leading them toward the editing systems for refolding or degradation. Therefore, UPR is an important mechanism in the ER, which is responsible for determining the fate of proteins, leading the cell toward the apoptosis or survival. ROS production and ER stress activate three arms of UPR in the cell. Among them, activation of two arms, IRE1α (Inositol Requiring Enzyme1) and PERK (PKR-like Endoplasmic Reticulum Kinase), result in the splicing of XBP1 and phosphorylation of eIf2α, respectively, which ultimately led to cell apoptosis [49, 50]. While, the ATF6 pathway slows the pace of protein translation, reduces the load of protein into the ER, and induces the production of molecular chaperones.

Since the obtained data in this study confirmed HG-induced ROS and NO production and apoptosis induction in the studied cells, we investigated two processes of XBP1 splicing and eIf2α phosphorylation as the possible mechanisms of apoptosis induction through UPR.

As shown in the results, the ratio of XBP1s/XBP1u (in both mRNA and protein levels) and the ratio of p-eIf2α/eIf2α were significantly increased in the HG-treated C2C12 myotubes and 3T3-L1 adipocytes, suggesting that ER stress induced by HG may lead the cells to apoptosis. Both of these markers were significantly decreased due to the Lys treatment. Previous studies have shown the role of Lys as an antioxidant that upregulates the anti-inflammatory factors [19, 45] and reduces the glycoxidation markers in rat model of diabetes-atherosclerosis [51].

eIF2α has been known to play a very important role in the UPR-induced autophagy [52]. Due to the UPR activation, phosphorylated form of eIF2α (*p*-eIF2α) decreases general protein synthesis and permits the transcription of genes involved in autophagy and apoptosis [53]. Thus, *p*-eIF2α was investigated as one of the ER stress sensors [54, 55].

Autophagy is a complex mechanism that under specific circumstances follows a "cell survival" or a "cell-killing" strategy. Actually, autophagy involves the sequestration of damaged organelles and misfolded/aggregated proteins [56, 57]. The conversion of LC3 I to LC3II, the cytosol to membrane transition and accumulation of LC3II protein has been introduced as an autophagy marker [58, 59]. Previous studies indicated a relation and crosstalk between UPR, autophagy and apoptosis in cancer cells [60]. So that, one of the mechanisms involved in the ER homeostasis regeneration and UPR modification is autophagy stimulation. In addition, persistent or sever ER stress can shift the cytoprotective functions of both UPR and autophagy into cell death promoting mechanisms [61].

Activation of autophagic pathways has been shown in mouse embryos and oocytes in response to a hyperglycemic environment [12]. Sato *et al.* have also shown that 10 mM Lys decreased the ratio of LC3II/LC3I in C2C12 myotubes and inhibited autophagic–lysosomal system activity [62]. Here, investigation of the LC3 accumulation indicated that following treatment of the cells with HG, the level of LC3II and the LC3II/LC3I ratio was significantly increased; these are implying the autophagy activation in both myotubes and adipocytes. These data accompanying with the data about the elevation of apoptosis in these cells, confirming the role of autophagy-induced apoptosis. Lys treatment decreased significantly the LC3II/LC3I ratio in both cells, indicating the inhibitory role of Lys in this pathway.

Sato *et al.* has shown that 5–10 mM Lys stimulated the rate of protein synthesis in C2C12 myotubes and introduced Lys as a regulator of protein synthesis [63]. Here, we showed the inhibition of autophagy by 1 mM Lys. It means that Lys at lower concentrations, at least, prevents protein degradation through autophagy. This effect was mostly shown at the first hours of HG treatment.

Finally, Lys significantly increased the cell survival and decreased the HG-induced apoptosis after 6 h of treatment. But, there were significant differences between these cells and control cells. It is possible that by increasing the time course of the experiment to 12 or 24 h these improvement effect is intensified. The subject that should be studied in the near future. The results of the present study are consistent with the previously reported data and indicated that Lys acts through different mechanisms. Like a chemical chaperone, Lys inhibits protein glycation and misfolding; and like an antioxidant, inhibits the oxidative stress. In addition, it acts as a regulator of UPR and autophagy, and controls the protein turnover in the cells. Further studies are needed to evaluate these mechanisms in the *in vivo*, and in different tissues.

In conclusion, the presented data indicated that Lys, as an antioxidant and a regulator of UPR and autophagy, protects C2C12 myotubes and 3T3-L1 adipocytes against damages induced by HG.

## Supporting information

**S1 Fig. Representative images of C2C12 cells.** (A) in control medium and (B) in differentiation medium after 6 days.
(TIF)

**S2 Fig. Oil Red O staining of 3T3-L1 cells.** (A) The 3T3-L1 cells staining at first, and (B) after 2 weeks incubation in differentiation media. A significant increase in the lipid droplets stained with Oil Red O, in the peri-nuclear region, indicating the differentiation of 3T3-L1 cells to 3T3-L1 adipocytes. The white marker on the right bottom of the figures indicating a 25 μm scale.
(TIF)

**S1 File. The raw images of all PCR and Western blot data.** Each figure was named according to the related figure in the text.
(PDF)

## Acknowledgments

The Research Council of Tarbiat Modares University paid for preparation of materials in this project. It had no role in study design, data collection and analysis, decision to publish, or preparation of the manuscript. We also gratefully acknowledge Mr. Ahmad Nasimian for some technical assistance.

## Author Contributions

**Conceptualization:** S. Mehdi Ebrahimi, S. Zahra Bathaie, Mohammad Taghikhani, Manouchehr Nakhjavani.

**Data curation:** S. Mehdi Ebrahimi, Nassim Faridi.

**Formal analysis:** S. Mehdi Ebrahimi, S. Zahra Bathaie, Nassim Faridi, Soghrat Faghihzadeh.

**Funding acquisition:** S. Zahra Bathaie, Manouchehr Nakhjavani.

**Investigation:** S. Mehdi Ebrahimi, S. Zahra Bathaie.

**Methodology:** S. Mehdi Ebrahimi, Soghrat Faghihzadeh.

**Project administration:** S. Zahra Bathaie, Manouchehr Nakhjavani.

**Resources:** S. Zahra Bathaie.

**Supervision:** S. Zahra Bathaie, Mohammad Taghikhani, Manouchehr Nakhjavani.

**Validation:** S. Zahra Bathaie, Manouchehr Nakhjavani, Soghrat Faghihzadeh.

**Visualization:** S. Zahra Bathaie, Nassim Faridi.

**Writing – original draft:** S. Mehdi Ebrahimi, S. Zahra Bathaie, Nassim Faridi, Mohammad Taghikhani.

**Writing – review & editing:** S. Zahra Bathaie, Manouchehr Nakhjavani.

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
