## [Decision Letter · Decision Letter 0]

14 Aug 2019

PONE-D-19-20629

L-Lysine Protects C2C12 and 3T3-L1 Cells Against High Glucose Damages and Stresses

PLOS ONE

Dear Professor Bathaie,

Thank you for submitting your manuscript to PLOS ONE. After careful consideration, we feel that it has merit but does not fully meet PLOS ONE’s publication criteria as it currently stands. Therefore, we invite you to submit a revised version of the manuscript that addresses the points raised during the review process.

Your manuscript was reviewed by two knowledgeable referees in this area, and their comments are appended. As you will see, they both had numerous concerns that will need to be addressed by the authors before I can proceed further. The authors need to properly addressed all these concerns to satisfy our reviewers.

We would appreciate receiving your revised manuscript by Sep 28 2019 11:59PM. To enhance the reproducibility of your results, we recommend that if applicable you deposit your laboratory protocols in protocols.io, where a protocol can be assigned its own identifier (DOI) such that it can be cited independently in the future. For instructions see: http://journals.plos.org/plosone/s/submission-guidelines#loc-laboratory-protocols

We look forward to receiving your revised manuscript.

Kind regards,

Makoto Kanzaki, Ph.D.

Academic Editor

PLOS ONE

Journal Requirements:

1.

PLOS ONE now requires that authors provide the original uncropped and unadjusted images underlying all blot or gel results reported in a submission’s figures or Supporting Information files. This policy and the journal’s other requirements for blot/gel reporting and figure preparation are described in detail at https://journals.plos.org/plosone/s/figures#loc-blot-and-gel-reporting-requirements and https://journals.plos.org/plosone/s/figures#loc-preparing-figures-from-image-files. When you submit your revised manuscript, please ensure that your figures adhere fully to these guidelines and provide the original underlying images for all blot or gel data reported in your submission. See the following link for instructions on providing the original image data: https://journals.plos.org/plosone/s/figures#loc-original-images-for-blots-and-gels.

Reviewers' comments:

Reviewer's Responses to Questions

**Comments to the Author**

1. Is the manuscript technically sound, and do the data support the conclusions?

Reviewer #1: Yes

Reviewer #2: Partly

2. Has the statistical analysis been performed appropriately and rigorously? 

Reviewer #1: Yes

Reviewer #2: Yes

3. Have the authors made all data underlying the findings in their manuscript fully available?

Reviewer #1: Yes

Reviewer #2: Yes

4. Is the manuscript presented in an intelligible fashion and written in standard English?

Reviewer #1: Yes

Reviewer #2: Yes

5. Review Comments to the Author

Reviewer #1: 1.Fig.1, the CK specific activity was significantly increased in C2C12 cells after 6 days. Whether there is statistical difference between the two groups, if the difference is statistically significant, it should be labeled.

2.Fig.3A, the results showed that Lys was not toxic to C2C12 myotubes and 3T3-L1 adipocytes at different concentrations (0.5 to 10 mM). Therefore, the authors used 1 mM Lys in other experiments. The authors do not show the reason for choosing the concentration of 1 mM，this is confusing for readers to understand.

3."Figs. 3B and 3C show the effect of HG and HG+Lys, respectively, on viability of both C2C12 myotubes and 3T3-L1 adipocytes at different time courses (2, 4 and 6 h).These figures indicate a time dependent and significant decreases (p= 0.000) in the cell viability due to the HG treatment in comparison with the control cells. Lys treatment significantly overcomes the HG toxicity. So that, the p values of the differences in the cell viabilities between groups treated by Lys+HG and groups treated with HG were p= 0.013 C2C12 myotubes and were p= 0.008 in 3T3-L1 adipocytes. However, there are still significant differences between Lys+HG groups and control groups in both cell lines (p= 0.000), which possibly compensated by increasing time up to 24 h." There are statistically significant differences between the groups, the authors should be labeled p value in the figure 3 B and C.

4.Fig.5A and 5B, in the result, the author’s description is that " The repeated measurement shows significant increase (p=0.000) in the ROS production in HG groups in comparison with the controls of both cell types. " and "Due to Lys treatment, significant decrease (p=0.000) in the ROS levels was observed in HG+Lys groups in comparison with HG groups in both C2C12 and 3T3-L1 cells. " However, the labeled in the figure does not match with the result description.

5.Fig.5Cand 5D, the labeled in the figure does not match with the result description.

6.Fig.6Cand 6D, in the result, the author’s description is that “indicated the significant (p=0.000) increase in the XBP1s in HG groups in comparison with the control. This value (the spliced form of XBP1 mRNA) was significantly decreased (p=0.000) after Lys treatment in the HG+Lys groups of both cells.” However, the labeled in the figure does not match with the result description.

7.Fig. 7B,C,D and 8B,C,D, the labeled in the figure does not match with the result description.

8.In Fig. 7A and 8A, labeled is p-eIF2α, but in Fig. 7C and 8C, labeled is eIF2α.

9.What does that mean "a,b,c,d,e and f " in Table 1?

10.The authors should proofread for grammatical and punctuation.

Reviewer #2: The authors investigated the effect of L-Lysine (Lys) on high glucose induce damages in C2C12 and 3T3-L1 cells. They suggested that the Lys prevents HG-induced detrimental effects of following parameters ROS, NO, apoptosis, a spliced form of XBP1, p-elf2a, and LC3II/LC3I ratio. However, although the findings are valuable, they are not well described or discussed and even overstated or misinterpreted at times. There are many other minor errors of syntax and grammar throughout the text, which need to be fixed. The manuscript needs to remove the multiple inaccuracies and inconsistencies, improve the logical flow, and reformat for better clarity. There are some suggestions and concerns which should be addressed.

Major comments:

1. Abstract: Must be rewritten within the scope of the study. Without any rationale given for the parameters studied, the abstract creates an impression that the study describes random outcomes of random in vitro treatments. Clear hypothesis and logical explanation of the results are often lacking throughout the paper, although they are essential for the scientific method. Conclusion: “These results provide new insights…..” does not make logical sense.

2. Introduction last paragraph: (lines 73-79): “Continuing with our previous studies,………” it means reference [24, 25] and those studies conducted on fibrogenic activity (hyperglycemia) and lysosome glycation in diabetic condition and protective effect of Lys. In what these studies continuing with the previous observations apart from hyperglycemia or diabetic condition, neither fibrosis or lysosome glycation studied in the current manuscript. Avoid misinterpret and overstate, strict with the scope of the study.

3. Materials and methods: Justification must be included in materials and methods for the treatment doses used (Lys 1mM) and even though cells not shown any cytotoxic effect with a range between 0.5-10mM (supraphysiologic) Lys in both cells; it is not around physiology Lys levels?. Moreover, the rationale for Lys instead of well-known Cys and Gly (precursor for major antioxidant Glutathione).

Line 136: Is the percentage of viable cells out of total cells counted? Please specify.

Line 148:” The obtained raw results were analyzed by FlowJo software”….Sentence not completed…How it expressed? Mention that on the next line — same applicable to “Measuring the ROS and NO production” also.

Line 163-180: What was the internal control used for RT-PCR and how it normalized? “No loading control in gels ?”

Line 183: “The expression levels of elf2a…..” elf2a antibody not mentioned in materials method only phosphorylated form mentioned and in blot also p-elf2a mentioned but quantified bar diagram for elf2a. Why this error all over the manuscript?

Line 201-207: Statistical Analysis: What Post Hoc analysis used?

4. Subtitles for Results section poor: say for example, “Cell culture and differentiation, Cell viability Assay, Apoptosis detection, ….The results of RT-PCR and Western blotting” seems methods title rewrite according to the observed parameter for results. The results have to precisely and entirely describe the findings in the figures.

5. Beyond ROS and NO assays; the essential relevant hydrogen peroxide assay, protein carbonyl, and GSH or total antioxidant capacity are of remarkable to validate the oxidative damage point of the study. It is interesting to see those data in the present study; the authors may include those parameters to validate the present observations.

6. Discussion: The discussion as a whole lacks logical clarity and needs to be rewritten to discuss how the described findings fit in the hypothesis and what their implications for the future are?.

Line: 362-369 “This paragraph consists of what seems like random facts without a logical connection between them.” Apoptosis and autophagy are two different opposite entity; how could authors explain this contradictory in their study? Justification need and discussion must be included in this aspect.

“Line 323….the pseudo-diabetic condition” What was that? What it mean?.

7. The title, aim of the study, abstract conclusion, introduction conclusion, and looks different and often overstated than what was observed. Line:375-378. The conclusion is overstated in discussion “As the presented data here………..induced by HG” This is purely an in vitro study on two cell lines and few parameters studies to elucidate the beneficial role of Lys on HG-induced damages. The conclusion should be drawn from the significant outcome of the study. Hence, the conclusion must be rewritten more precisely with the outcome from the observations. “Correct the conclusion points in the abstract and introduction also.” It applies to the title; it is better to avoid overstating.

8. Legends often confusing must be rewritten.

8. Multiple inaccuracies in the Figures:

Fig 1 and 2: Is not a novel one, it has been shown many times in literature? Moreover, myotubes images may be included, and oil o red must be quantified. Moreover, suggestive of combining these known results and may be provided as supplementary.

Fig 3. Significant missing on the top of the bar diagram. Sub figure a, b, c marking also missing and it is good to mention an insert whether it is C2C12 or 3T3-L1 data inside in all figures.

Fig 4. Repetition of Table 1. Either one should move to supplementary data. Moreover, what is replicated for the flow cytometry apoptosis experiment?

Fig 5. How the groups are compared only between time points not between groups? It is better to analyze two-way ANOVA for analysis of different time points and groups.

Fig 6. It was early mentioned that Gels lacks internal controls. This is not acceptable.

Fig 7 and 8. Western blot bands are not representative; provide representative blot images and bands seem oversaturated; it is advisable to reduce the loading protein concentration.

Why different groups cropped separately whether those samples run on different gels separately?

p-elf2a must be normalized with elf2a, not with b-actin. Check the axis for p-elf2a it was mentioned elf2a/b-action. This is not acceptable.

Poor figure resolution was challenging to see data and labels.

Other:

Provide correct company name, place, and catalog for all chemicals and assay kits.

Even though the scale bar is presented in figures, it is good to provide magnification in legends along with scale bar for all microscope images.

Along with CK activity; Why not myotubes images provided? It would be interesting similar like differentiated adipocytes.

PCR and blotting experiments are three independent biological replicates, and what was the technical replicate?

All the abbreviations have to be spelled out in the abstract. In the main body of the manuscript, the full name and abbreviation of the name have to be spelled out on the first mention only, with just the abbreviated form used for the rest of the text. The manuscript is peppered with many abbreviations changing back and forth between their full and abbreviated versions throughout the text, which makes reading confusing at times. Why glucose abbreviated as Glc? Avoid non-standard abbreviations throughout the manuscript.

Please double-check grammar and English in the manuscript and check for typo errors. These must be avoided throughout the manuscript.

6. PLOS authors have the option to publish the peer review history of their article (what does this mean?). If published, this will include your full peer review and any attached files.

Reviewer #1: No

Reviewer #2: No

---

## [Author Response · Author response to Decision Letter 0]

2 Oct 2019

Responses to the Reviewers' comments:

Thank you very much for your kind considerations, positive view to the manuscript and very good comments.

5. Review Comments to the Author

We would like to say thank you very much to both of the reviewers for their time and constructive comments. We understand and sympathize with the request for more details from the reviewers.

Below we addressed all comments and suggestions (reviewer’s comments in black, our reply in blue). We have considered the various suggestions made by the reviewers and have accordingly revised and rewritten the manuscript. Hopefully it is fulfilled the editor and reviewers’ expectations and be accepted for publication.

 

Reviewer #1:

1.Fig.1, the CK specific activity was significantly increased in C2C12 cells after 6 days. Whether there is statistical difference between the two groups, if the difference is statistically significant, it should be labeled.

Yes, you are right. The scale bar and the statistical difference was shown in the figure. Please look at Line 220.

2.Fig.3A, the results showed that Lys was not toxic to C2C12 myotubes and 3T3-L1 adipocytes at different concentrations (0.5 to 10 mM). Therefore, the authors used 1 mM Lys in other experiments. The authors do not show the reason for choosing the concentration of 1 mM, this is confusing for readers to understand.

Physiologic concentration of Lys has been reported about 232 µM (European Journal of Clinical Nutrition (2016) 70, 306–312) or 92 ± 6 µM (Boyvin et al., 2017). However, determination of the ratio of LC3-II/I in C2C12 myotubes has been shown that 10 mM Lys inhibited autophagic–lysosomal system activity (Sato et al., 2014). Then, the authors showed that Lys regulates protein synthesis and at 5 – 10 mM stimulated the rate of protein synthesis in C2C12 myotubes (Sato et al., 2016). High Lys concentration has also shown toxicity against HK-2 kidney cells (Verzola et al., 2012). These explanations and references were added in the text.

In the present study, although Lys showed similar viability at concentrations 1 and 5 mM, and even at 10 mM, for avoiding the above mentioned side effects and toxicity, we used the minimum pharmacologic concentration, in all other experiments. Please look at Lines 321-327.

3."Figs. 3B and 3C show the effect of HG and HG+Lys, respectively, on viability of both C2C12 myotubes and 3T3-L1 adipocytes at different time courses (2, 4 and 6 h). These figures indicate a time dependent and significant decreases (p= 0.000) in the cell viability due to the HG treatment in comparison with the control cells. Lys treatment significantly overcomes the HG toxicity. So that, the p values of the differences in the cell viabilities between groups treated by Lys+HG and groups treated with HG were p= 0.013 C2C12 myotubes and were p= 0.008 in 3T3-L1 adipocytes. However, there are still significant differences between Lys+HG groups and control groups in both cell lines (p= 0.000), which possibly compensated by increasing time up to 24 h." There are statistically significant differences between the groups; the authors should be labeled p value in the figure 3 B and C.

The mentioned Figs. that converted to Figures 2B and 2C in the revised version, were labeled. According to your comments about other figures (4 to 8 of the previous version), we revisited the statistical analysis with the help of Professor Faghihzadeh and all parts was rewritten for more clarity.

4.Fig. 5A and 5B, in the result, the author’s description is that " The repeated measurement shows significant increase (p=0.000) in the ROS production in HG groups in comparison with the controls of both cell types. " and "Due to Lys treatment, significant decrease (p=0.000) in the ROS levels was observed in HG+Lys groups in comparison with HG groups in both C2C12 and 3T3-L1 cells. " However, the labeled in the figure does not match with the result description.

5.Fig. 5C and 5D, the labeled in the figure does not match with the result description.

6.Fig. 6C and 6D, in the result, the author’s description is that “indicated the significant (p=0.000) increase in the XBP1s in HG groups in comparison with the control. This value (the spliced form of XBP1 mRNA) was significantly decreased (p=0.000) after Lys treatment in the HG+Lys groups of both cells.” However, the labeled in the figure does not match with the result description.

7.Fig. 7B,C,D and 8B,C,D, the labeled in the figure does not match with the result description.

In response to the comments 4 to 7, we should say that, we used repeated-measures ANOVA for statistical analysis of the differences between different groups (C, HG, HG+Lys and Lys) at different times (0, 2, 4 and 6 h). In the previous version we wrote these results separately in the results and Figs. But, for answering your valuable questions and for more clarity of the statistical analysis and data, we asked Prof. Faghihzadeh to help us and we added his name in the paper. Please note to the new analysis and writing of the statistics in the results and all Figures.

8.In Fig. 7A and 8A, labeled is p-eIF2α, but in Fig. 7C and 8C, labeled is eIF2α.

You are right. It was our fault. We edited it and added the data of eIF2α. Thus, the p-eIF2α/ eIF2α ratio at different treatment condition was added in the results.

9.What does that mean "a,b,c,d,e and f " in Table 1?

Since all the data were statistically significant, p=0.000, we deleted them.

10.The authors should proofread for grammatical and punctuation.

The proofreading was performed in the manuscript and we tried to edit all the writing errors. Hope it is acceptable, now.

 

Reviewer #2:

The authors investigated the effect of L-Lysine (Lys) on high glucose induce damages in C2C12 and 3T3-L1 cells. They suggested that the Lys prevents HG-induced detrimental effects of following parameters ROS, NO, apoptosis, a spliced form of XBP1, p-elf2a, and LC3II/LC3I ratio. However, although the findings are valuable, they are not well described or discussed and even overstated or misinterpreted at times. There are many other minor errors of syntax and grammar throughout the text, which need to be fixed. The manuscript needs to remove the multiple inaccuracies and inconsistencies, improve the logical flow, and reformat for better clarity. There are some suggestions and concerns which should be addressed. 

Thank you very much for your kind consideration and good suggestions. As you suggested, grammatical and syntax errors was carefully checked, revised and improved. We hope the revised manuscript meets your expectations.

Major comments:

1. Abstract: Must be rewritten within the scope of the study. Without any rationale given for the parameters studied, the abstract creates an impression that the study describes random outcomes of random in vitro treatments. Clear hypothesis and logical explanation of the results are often lacking throughout the paper, although they are essential for the scientific method. Conclusion: “These results provide new insights…..” does not make logical sense.

The Abstract was revised and remodeled. We tried to include the hypothesis and logical explanation of the results.

2. Introduction last paragraph: (lines 73-79): “Continuing with our previous studies,………” it means reference [24, 25] and those studies conducted on fibrogenic activity (hyperglycemia) and lysosome glycation in diabetic condition and protective effect of Lys. In what these studies continuing with the previous observations apart from hyperglycemia or diabetic condition, neither fibrosis or lysosome glycation studied in the current manuscript. Avoid misinterpret and overstate, strict with the scope of the study.

You are right. Although in the present study we have not investigated all the processes we investigated previously, one by one, we believe that stress induction by HG has an important role in all of the diabetic complications and we focused here on some markers indicating the stress induction by HG and suppressive effect of Lys, in these two cell lines. As you suggested, we have corrected the sentence.

3. Materials and methods: Justification must be included in materials and methods for the treatment doses used (Lys 1mM) and even though cells not shown any cytotoxic effect with a range between 0.5-10mM (supraphysiologic) Lys in both cells; it is not around physiology Lys levels?. 

Physiological level of Lys in the serum is about 232 µM (European Journal of Clinical Nutrition (2016) 70, 306–312) or 92 ± 6 µM (Boyvin et al., 2017). Thus, we used higher concentrations in this study. Sato et al. used 0.1 to 10 mM of Lys for C2C12 myotubes (Bioscience, Biotechnology, and Biochemistry, 2016 Vol. 80, No. 11, 2168–2175).

Furthermore, 5- 10 mM of Lys induced the protein synthesis, and Lys 10 mM has been introduced as a potent suppressor of autophagy, while Lys 0.1 mM had no effect on autophagy (Bioscience, Biotechnology, and Biochemistry, 2016 Vol. 80, No. 11, 2168–2175).

The toxicity of 5, 10 and 20 mM of Lys against cancer cell through induction of H2O2 production has been also shown (J. Phys. D: Appl. Phys. 49 (2016) 274001 (7pp)).

A decrease in HK-2 cell viability has been reported with high Lysine (10-15 mM) concentration (J Inherit Metab Dis., DOI 10.1007/s10545-012-9468-z).

However, as shown in Figs.2 A & B, Lys showed a slight toxicity against both cells at 10 mM up to 6 h, which may increase by time. Since there was no autophagic induction in very low Lys concentration (0.1 mM) and we would like to suppress HG-induced autophagy in the cells, with no more toxicity, we used 1 mM in all other experiments.

These explanations and references were added in the text.

Moreover, the rationale for Lys instead of well-known Cys and Gly (precursor for major antioxidant Glutathione).

We used previously several amino acids in our studies (References 1 & 2). Although all of them was effective, we obtained the best values for Lys in the diabetes treatment. Thus, in the present study we applied Lys.

Line 136: Is the percentage of viable cells out of total cells counted? Please specify.

Yes, the viable cells in the control was assumed as 100 and the viability in the treated cells was calculated relative to the control. It was explained in the text (Lines 143-144).

Line 148:” The obtained raw results were analyzed by FlowJo software”….Sentence not completed…How it expressed? Mention that on the next line — same applicable to “Measuring the ROS and NO production” also.

Thanks for your careful reading the manuscript and valuable comments. They were completed Please look at Lines 155-156 for Software, Lines 163-164 for ROS and Lines 169-171 for NO.

Line 163-180: What was the internal control used for RT-PCR and how it normalized? “No loading control in gels?”

We measured RNA concentrations of samples with a NanoDrop device. In addition, the integrity and purity of the RNA samples was evaluated via ethidium bromide visualization of intact 18S and 28S RNA bands after agarose gel electrophoresis. 

For normalization, the same cDNA concentration of different samples was determined by NanoDrop. 

The internal control, HPRT, was also used but, we did not report the data. Since our goal was showing the splicing of the XBP1 gene, and compare it with the unspliced form, we did not report the data of HPRT. Please note that both of the spliced and unspliced forms, if exist, are seeing in one run and in a gel. Other references in the literature that have not used internal control are include: (J Clin Invest. 2008;118(6):2148-2156. https://doi.org/10.1172/JCI33777) and (Blood, 2012, 119:5772-5781; doi: https://doi.org/10.1182/blood-2011-07-366633).

However, if you think we must report it, we will do.

Line 183: “The expression levels of elf2a…..” elf2a antibody not mentioned in materials method only phosphorylated form mentioned and in blot also p-elf2a mentioned but quantified bar diagram for elf2a. Why this error all over the manuscript?

We apologize for the confusion and thank you for pointing out this problem. We added the data of elf2a in the revised version and the sentence in the paper was corrected accordingly. Please look at Line 100.

Line 201-207: Statistical Analysis: What Post Hoc analysis used?

We used Tukey analysis. It was added in the manuscript. Please look at Line 212.

4. Subtitles for Results section poor: say for example, “Cell culture and differentiation, Cell viability Assay, Apoptosis detection, ….The results of RT-PCR and Western blotting” seems methods title rewrite according to the observed parameter for results. The results have to precisely and entirely describe the findings in the figures.

Thank you for the great suggestions. Because of your suggestion, we changed the subtitles that precisely describe the findings in each section.

5. Beyond ROS and NO assays; the essential relevant hydrogen peroxide assay, protein carbonyl, and GSH or total antioxidant capacity are of remarkable to validate the oxidative damage point of the study. It is interesting to see those data in the present study; the authors may include those parameters to validate the present observations.

Yes, you are right. But in the present study our main goal was ERS and autophagy, we focused on the mentioned parameters. Since other parameters was extensively reported in the literature and even by us, we only examined and reported two parameters (ROS and NO) for confirmation and other reports were referenced.

6. Discussion: The discussion as a whole lacks logical clarity and needs to be rewritten to discuss how the described findings fit in the hypothesis and what their implications for the future are?

Thank you very much for your advice. The discussion has rewritten and revised to clarify the logical processes. We have considered your suggestions and have accordingly rewritten the manuscript to describe findings fit in the hypothesis and their implications for the future.

Line: 362-369 “This paragraph consists of what seems like random facts without a logical connection between them.” Apoptosis and autophagy are two different opposite entity; how could authors explain this contradictory in their study? Justification need and discussion must be included in this aspect.

Autophagy is a complex mechanism that under specific circumstances follows a “cell survival” or a “cell-killing” strategy. Actually, autophagy involves the sequestration of damaged organelles and misfolded/aggregated proteins. Some more explanation was added in Lines 386-397 of the revised manuscript.

“Line 323….the pseudo-diabetic condition” What was that? What it mean?

Thank you for pointing out that mistake. It was not suitable here and we replaced it with “in vitro model of diabetes”. Please look at Line 346.

7. The title, aim of the study, abstract conclusion, introduction conclusion, and looks different and often overstated than what was observed. Line:375-378. The conclusion is overstated in discussion “As the presented data here………..induced by HG” This is purely an in vitro study on two cell lines and few parameters studies to elucidate the beneficial role of Lys on HG-induced damages. The conclusion should be drawn from the significant outcome of the study. Hence, the conclusion must be rewritten more precisely with the outcome from the observations. “Correct the conclusion points in the abstract and introduction also.” It applies to the title; it is better to avoid overstating.

We have revised and rewritten the conclusion points of abstract and introduction section by adding and deleting some sentences.

8. Legends often confusing must be rewritten.

Yes, you are right. Please accept our apology. All of them were amended.

8. Multiple inaccuracies in the Figures:

Fig 1 and 2: Is not a novel one, it has been shown many times in literature? Moreover, myotubes images may be included, and oil o red must be quantified. Moreover, suggestive of combining these known results and may be provided as supplementary.

Yes, you are right. We omitted these figures from the manuscript.

Fig 3. Significant missing on the top of the bar diagram. Sub figure a, b, c marking also missing and it is good to mention an insert whether it is C2C12 or 3T3-L1 data inside in all figures. 

Thank you. All of them were edited.

Fig 4. Repetition of Table 1. Either one should move to supplementary data. Moreover, what is replicated for the flow cytometry apoptosis experiment?

Figure show the percentages of four different stages of the cells, but in Table we presented the overall percentage of apoptosis, i.e. both early and late apoptosis together. We think it helps to better understanding the situation.

Fig 5. How the groups are compared only between time points not between groups? It is better to analyze two-way ANOVA for analysis of different time points and groups.

Actually, the two-way ANOVA compares the mean differences between groups with two independent variables. On the other word, a two-way ANOVA is designed to assess the interrelationship of two independent variables on a dependent variable.

In our study, time dependency of changes is not an independent variable. Thus, we used repeated-measures ANOVA for statistical analysis of the differences between different groups (C, HG, HG+Lys and Lys) at different times (0, 2, 4 and 6 h). In the previous version we wrote these results separately in the results and Figures. But, for answering your valuable question and for more clarity of the statistical analysis and data, we asked Prof. Faghihzadeh to help us and we added his name in the paper. Please note to the new analysis and writing the statistics.

Fig 6. It was early mentioned that Gels lacks internal controls. This is not acceptable.

As responded to the above mentioned comments about Lines 163-180, our goal was showing the induction of XBP1 splicing and the ratio of spliced form to the unspliced form was important for us. The same as you expected for p-eIf2α and its ration against the unphosphorylated form.

Fig 7 and 8. Western blot bands are not representative; provide representative blot images and bands seem oversaturated; it is advisable to reduce the loading protein concentration.

You are right; since we repeated the experiment several times, other representative images of western blot were replaced.

Why different groups cropped separately whether those samples run on different gels separately?

Yes, we have electrophoresis system (Mini-PROTEAN Tetra Cell of Bio-Rad) with a 10-well system; but, we had 12 samples (C, HG, HG+Lys and Lys at different times of 2, 4 and 6 h). Thus, we run C, HG and HG+Lys groups in one gel and Lys was run on another gel.

p-elf2a must be normalized with elf2a, not with b-actin. Check the axis for p-elf2a it was mentioned elf2a/b-action. This is not acceptable. 

Yes, you are right. It was our fault. We included the data of elf2α and the data of p-elf2α was normalized with it.

Poor figure resolution was challenging to see data and labels.

We apologize for the poor figure resolution. We have improved it in the revised paper

Other:

Provide correct company name, place, and catalog for all chemicals and assay kits.

The company name, place, and catalog number for all chemicals and assay kits was included in the text. Please look at Lines 88-103.

Even though the scale bar is presented in figures, it is good to provide magnification in legends along with scale bar for all microscope images.

The magnification of scale bar for all microscope images was included in legends.

Along with CK activity; Why not myotubes images provided? It would be interesting similar like differentiated adipocytes.

Please look at the response to comment 8 and Fig. 1S. As reviewer suggested the myotubes images was included.

PCR and blotting experiments are three independent biological replicates, and what was the technical replicate?

All PCR and blotting experiments were also technically duplicate or triplicate.

All the abbreviations have to be spelled out in the abstract. In the main body of the manuscript, the full name and abbreviation of the name have to be spelled out on the first mention only, with just the abbreviated form used for the rest of the text. The manuscript is peppered with many abbreviations changing back and forth between their full and abbreviated versions throughout the text, which makes reading confusing at times. Why glucose abbreviated as Glc? Avoid non-standard abbreviations throughout the manuscript.

All the abbreviations were spelled out in the abstract and in the text at first time using.

https://www.researchgate.net/figure/Abbreviations-Glc-glucose-GlcA-glucuronic-acid-Gal-galactose-Ac-acetyl_fig1_266325670

glucose abbreviation glc

https://www.google.com/search?rlz=1C1GCEA_enUS863US863&sxsrf=ACYBGNQ1kKFBmrcPS8KCTG1tcNwUolLHBQ:1569501820772&q=glucose+abbreviation+glc&sa=X&ved=2ahUKEwi-jYH0we7kAhWJy6QKHSYoB0UQ1QIoAHoECA0QAQ

https://www.ccrc.uga.edu/~rcarlson/SugAbr.pdf

http://www.jbc.org/site/misc/abbrev.xhtml

GlcNAc N-acetylglucosamine

As it is seen in the above mentioned Web Sites, Glc has been introduced as the abbreviation of glucose. In chemistry, it is usually used and even JBC (Journal of Biological Chemistry) has been introduced GlcNAc. In addition, we defined it in both abstract and throughout the text. However, if you think it is not standard, we can change it to glucose in all the text.

Please double-check grammar and English in the manuscript and check for typo errors. These must be avoided throughout the manuscript.

We did it again and hope that it is acceptable.

References: 

1) F. Bahmani, S.Z. Bathaie, S.J. Aldavood, A. Ghahghaei (2012). "Glycine therapy inhibits progression of cataract in streptozotocin-induced diabetic rats." Molecular Vision, 18, 439-448.

2) S. Mahdavifard, S.Z. Bathaie, M. Nakhjavani, H. Heidarzadeh (2014). “L-cysteine is a potent inhibitor of protein glycation on both albumin and LDL, and prevents the diabetic complications in diabetic-Atherosclerotic rat.” Food Research International, 96, 909-916.

---

## [Decision Letter · Decision Letter 1]

17 Oct 2019

PONE-D-19-20629R1

L-Lysine Protects C2C12 and 3T3-L1 Cells Against High Glucose Damages and Stresses

PLOS ONE

Dear Professor Bathaie,

Thank you for submitting your manuscript to PLOS ONE. After careful consideration, we feel that it has merit but does not fully meet PLOS ONE’s publication criteria as it currently stands. Therefore, we invite you to submit a revised version of the manuscript that addresses the points raised during the review process.

Your revised paper was reviewed by the original referees and their comments are appended. As you will see the reviewer #1 kindly raised several issues that will need to be properly addressed by the reviewers. The authors need to carefully address all his/her concerns to fully satisfy the reviewer.

We would appreciate receiving your revised manuscript by Dec 01 2019 11:59PM. To enhance the reproducibility of your results, we recommend that if applicable you deposit your laboratory protocols in protocols.io, where a protocol can be assigned its own identifier (DOI) such that it can be cited independently in the future. For instructions see: http://journals.plos.org/plosone/s/submission-guidelines#loc-laboratory-protocols

We look forward to receiving your revised manuscript.

Kind regards,

Makoto Kanzaki, Ph.D.

Academic Editor

PLOS ONE

Reviewers' comments:

Reviewer's Responses to Questions

**Comments to the Author**

1. If the authors have adequately addressed your comments raised in a previous round of review and you feel that this manuscript is now acceptable for publication, you may indicate that here to bypass the “Comments to the Author” section, enter your conflict of interest statement in the “Confidential to Editor” section, and submit your "Accept" recommendation.

Reviewer #1: (No Response)

Reviewer #2: All comments have been addressed

2. Is the manuscript technically sound, and do the data support the conclusions?

Reviewer #1: Yes

Reviewer #2: Yes

3. Has the statistical analysis been performed appropriately and rigorously? 

Reviewer #1: Yes

Reviewer #2: Yes

4. Have the authors made all data underlying the findings in their manuscript fully available?

Reviewer #1: Yes

Reviewer #2: Yes

5. Is the manuscript presented in an intelligible fashion and written in standard English?

Reviewer #1: Yes

Reviewer #2: Yes

6. Review Comments to the Author

Reviewer #1: This study investigated the effect of Lys on intracellular signaling pathways in the C2C12 myotubes and 3T3-L1 adipocyte. And found that Lys prevents the induced HG-cell death in both C2C12 myotubes and 3T3-L1 adipocytes through the reduction of oxidative stress and regulation of the UPR. I have some specific comments for the authors to consider in revising their manuscript.

1.”These figures indicate a significant decrease (p= 0.000) in the cell viability due to the HG treatment in comparison with the control cells.” and “Despite the improvement effect of Lys, the viability of the cells treated with Lys+HG was significantly (p= 0.000) lower than control cells, in both cell lines.” Which is” the control cells”, not explicitly stated in the manuscript. Figure 2 didn't show which were control cells.

2."Figures 3A and 3B show the percentages of C2C12 and 3T3-L1 cells, respectively, at each quarter." However, A and B are not marked in Figure 3.

3.In Figures 4 and 5, only showed the p-value of the difference between the time points, but the p-value of the difference between the groups is not labeled.

4.In Figures 6BC and 7BC, only showed the p-value of the difference between the time points, but the p-value of the difference between the groups is not labeled.

5.Western blot data should show loading control (β-Actin) in Figures 8 and 9.

6.In Figures 8A, molecular weight of LC3-I/LC3-II wrong labeled.

7.In Figures 8 and 9, only showed the p-value of the difference between the time points, but the p-value of the difference between the groups is not labeled.

8.Image and digital resolutions are unclear.

9.The authors should double-check grammatical and punctuation.

Reviewer #2: (No Response)

7. PLOS authors have the option to publish the peer review history of their article (what does this mean?). If published, this will include your full peer review and any attached files.

Reviewer #1: No

Reviewer #2: No

---

## [Author Response · Author response to Decision Letter 1]

26 Oct 2019

Reviewer #1: This study investigated the effect of Lys on intracellular signaling pathways in the C2C12 myotubes and 3T3-L1 adipocyte. And found that Lys prevents the induced HG-cell death in both C2C12 myotubes and 3T3-L1 adipocytes through the reduction of oxidative stress and regulation of the UPR. I have some specific comments for the authors to consider in revising their manuscript.

1.”These figures indicate a significant decrease (p= 0.000) in the cell viability due to the HG treatment in comparison with the control cells.” and “Despite the improvement effect of Lys, the viability of the cells treated with Lys+HG was significantly (p= 0.000) lower than control cells, in both cell lines.” Which is” the control cells”, not explicitly stated in the manuscript. Figure 2 didn't show which were control cells.

Thank you very much for your kind consideration and very useful comments. We included the explanation about the control in the legend of Figure. 1. Actually, time zero (0) was considered as control with no treatment.

2."Figures 3A and 3B show the percentages of C2C12 and 3T3-L1 cells, respectively, at each quarter." However, A and B are not marked in Figure 3.

You are right. I am so sorry. The labels were added to Figure 3.

3.In Figures 4 and 5, only showed the p-value of the difference between the time points, but the p-value of the difference between the groups is not labeled.

4.In Figures 6BC and 7BC, only showed the p-value of the difference between the time points, but the p-value of the difference between the groups is not labeled.

To avoid any confusion, we did not. But, as the respected reviewer suggested, Figures 4, 5, 6BC and 7BC were additionally labeled with stars; and the stars were defined in the legends.

5.Western blot data should show loading control (β-Actin) in Figures 8 and 9.

We added the loading control (β-Actin) in Figures 8 and 9.

6.In Figures 8A, molecular weight of LC3-I/LC3-II wrong labeled.

Thank you very much. It was corrected.

7.In Figures 8 and 9, only showed the p-value of the difference between the time points, but the p-value of the difference between the groups is not labeled.

As reviewer suggested, Figures 8 and 9 were additionally labeled with stars; and the stars were defined in the legends.

8.Image and digital resolutions are unclear.

They were clarified.

9.The authors should double-check grammatical and punctuation.

Thanks for your suggestions. The grammatical and punctuation mistakes in the manuscript have been carefully checked and corrected.

---

## [Decision Letter · Decision Letter 2]

15 Nov 2019

L-Lysine Protects C2C12 Myotubes and 3T3-L1 Adipocytes Against High Glucose Damages and Stresses

PONE-D-19-20629R2

Dear Dr. Bathaie,

We are pleased to inform you that your manuscript has been judged scientifically suitable for publication and will be formally accepted for publication once it complies with all outstanding technical requirements.

With kind regards,

Makoto Kanzaki, Ph.D.

Academic Editor

PLOS ONE

Additional Editor Comments (optional):

Reviewers' comments:

Reviewer's Responses to Questions

**Comments to the Author**

1. If the authors have adequately addressed your comments raised in a previous round of review and you feel that this manuscript is now acceptable for publication, you may indicate that here to bypass the “Comments to the Author” section, enter your conflict of interest statement in the “Confidential to Editor” section, and submit your "Accept" recommendation.

Reviewer #1: All comments have been addressed

2. Is the manuscript technically sound, and do the data support the conclusions?

Reviewer #1: Yes

3. Has the statistical analysis been performed appropriately and rigorously? 

Reviewer #1: Yes

4. Have the authors made all data underlying the findings in their manuscript fully available?

Reviewer #1: Yes

5. Is the manuscript presented in an intelligible fashion and written in standard English?

Reviewer #1: Yes

6. Review Comments to the Author

Reviewer #1: (No Response)

7. PLOS authors have the option to publish the peer review history of their article (what does this mean?). If published, this will include your full peer review and any attached files.

Reviewer #1: No

---

## [Editor Report · Acceptance letter]

12 Dec 2019

PONE-D-19-20629R2 

L-Lysine protects C2C12 myotubes and 3T3-L1 adipocytes against high glucose damages and stresses 

Dear Dr. Bathaie:

I am pleased to inform you that your manuscript has been deemed suitable for publication in PLOS ONE. Congratulations! Your manuscript is now with our production department. 

With kind regards,

on behalf of

Dr. Makoto Kanzaki 

Academic Editor

PLOS ONE